

# Modeling study of the impact of SO₂ volcanic passive emissions on the tropospheric sulfur budget

Claire Lamotte[1], Jonathan Guth[1], Virginie Marécal[1], and Martin Cussac[1]

[1]CNRM, Université de Toulouse, Météo-France, CNRS, Toulouse, France

**Correspondence:** Lamotte Claire (claire.lamotte@meteo.fr)

**Abstract.** The contribution of volcanic emissions is argued as non-linear on the sulfur species burden. Thus, well constraining volcanic emissions inventories is necessary to better study the impacts induced by these pollution sources on the troposheric sulfur composition, as well as on sulfur species concentrations and depositions at the global surface. In this paper, the changes induced by the update of the volcanic sulfur emissions inventory on the global chemistry-transport model MOCAGE (MOdèle de Chimie Atmosphérique à Grande Échelle) are studied. Unlike the current inventory [Andres and Kasgnoc (1998)], the new inventory [Carn et al. (2016, 2017)] includes contributions from both passive degassing and eruptive emissions with more accurate information. Eruptions are provided as daily total amounts of sulfur dioxide (SO₂) emitted by volcanoes, while degassing are provided as annual averages with annual uncertainties by volcanoes. Information on plumes altitudes is also available and has been used in the model. The choice is made to look at the year 2013, when a neglieable amount of eruptive volcanic SO₂ emissions is referenced, allowing us to focus the study on the impact of passive degassing emissions on the tropospheric sulfur budget. A validation against GOME-2 SO₂ total column and MODIS AOD observations shows the improvements of the model results with the new inventory. Because the global volcanic SO₂ flux changes from 13 Tg.yr$^{-1}$ in the current inventory to 23.6 Tg.yr$^{-1}$ in the new inventory, the updated inventory shows significant differences in the global sulfur budget, mainly in the free troposphere and in the tropics. Even though volcanic SO₂ emissions represent 15 % of the total annual sulfur emissions, the volcanic contribution to the tropospheric sulfate aerosol burden is 27 %. Moreover, a sensitivity study on passive degassing emissions, using the annual uncertainties of emissions per volcanoes, also confirmed the non-linear link between tropospheric sulfur species and volcanic emissions. This study highlights the necessity of estimates accurate volcanic sources in chemistry-transport models in order to properly simulate tropospheric sulfur species.

## 1 Introduction

Sulfur emissions come mainly from human activities (fossil fuel combustion) and natural volcanic activity [Andreae (1985)]. Among them, sulfur dioxide (SO₂) is a pollutant species, known to affect both human health and the environment. Because of their link to the formation of sulfate aerosols, acid rain and the variation of the climate [Chestnut (1995), Robock (2000, 2007), Smith et al. (2001), Schmidt et al. (2012)], SO₂ emissions had become a major concern in environmental policies, leading to strong reductions in anthropogenic emissions in recent decades. Thus, the relative proportion of volcanoes in the total sulfur emission sources tends to increase. Plus, Graf et al. (1997) concluded that the efficiency of volcanic emissions to contribute



to the tropospheric sulfate burden is greater than the efficiency of anthropogenic emissions, mostly because $SO_2$ lifetime is greater in altitude and therefore impacts longer and over larger areas. Thus, we now expect to detect more clearly changes in the global tropospheric sulfur species content due to spatio-temporal variations in volcanic sulfur emissions.

In order to better understand the processes leading to these variations in sulfur species budgets, the role of modelling is
important. And, of course, in models, it is necessary to well constrain volcanic sources. At the global scale, emission inventories (compilation of all available data on the globe) are used in models. Until recently, the most effective measurement instruments to assess volcanic emissions for building the inventories were the COSPEC ground-based instruments [Moffat and Millan (1971), Williams-Jones et al. (2008)] or one of the first satellite instruments: TOMS [Krueger et al. (1995), Seftor et al. (1997), Torres et al. (1998a, b)]. Andres and Kasgnoc (1998) used these instruments to create one of the first global inventories of
volcanic sulfur emissions. Furthermore, being compiled for the Global Emissions Inventory Activity (GEIA), it is the most widely used global dataset. For example, it has been implemented in several climate and chemistry-transport models [Chin et al. (2000), Liu et al. (2005), Shaffrey et al. (2009), Emmons et al. (2010), Lamarque et al. (2012), Savage et al. (2013), Walters et al. (2014), Michou et al. (2015)] and used in various studies on climate aerosol radiative forcing, ocean dimethyl sulfide (DMS) sensitivity or tropospheric aerosol budget [Adams and Seinfeld (2001), Takemura (2012), Michou et al. (2020),
Gondwe et al. (2003a, b), Gunson et al. (2006), Liu et al. (2007)]. Subsequently, other studies using the same techniques, or building on this first inventory by supplementing it with documented sets of sporadic eruptions, have provided further global inventories [Halmer et al. (2002), Diehl et al. (2012)].

But at the time these inventories were built, techniques for measuring emission fluxes were not very accurate in quantitative, spatial and temporal detection of volcanic sources. Indeed, ground-based instruments can only be used on easy-to-access
volcanoes and TOMS detection sensitivity was limited only to the largest eruptions. The available inventories were therefore incomplete. Andres and Kasgnoc (1998) work, with only one average value of all 25 years data measurements collected per volcano, reflects only a climatology without time variability. However, a lot of improvements have been made recently on satellite technologies, making possible to monitor volcanic emissions more accurately. The satellite global coverage enables to detect emission fluxes even from hard-to-access volcanoes. The accuracy of the measurements has also made possible to detect
not only the largest eruption fluxes but also smaller ones and persistent degassing [Yang et al. (2010), Thomas et al. (2011), Carn et al. (2013), Li et al. (2013)]. As well, thanks to newly developed algorithms, information on injection altitudes is available [Yang et al. (2009, 2010, 2013), Nowlan et al. (2011), Rix et al. (2012), Clarisse et al. (2014)], reducing the uncertainties of the characterization of volcanic sources. Ge et al. (2016) highlighted the improvements made on the sulfate direct radiative forcing using both eruptive and passive degassing data in a chemistry-transport model and stressed the importance of considering the
$SO_2$ injection altitude in volcanic emission inventories.

Carn et al. (2016, 2017), in its works, sought to compile all this new, more numerous and qualitative data in order to provide a more representative inventory of volcanic $SO_2$ emissions. It is a compilation of both eruptions and passive degassing at the global scale, providing data up to a daily frequency for eruptive emissions, and a yearly frequency along with uncertainty for passive source strength.





These new global volcanic sulfur inventories open the possibility of new, more detailed and accurate studies of the impact of volcanic emissions at the global scale; a huge change compared with last decades studies widely focused on major volcanic eruptions [Robock (2000)]. At the global scale, numerous studies aim at the assessment of the dispersion of sulfate aerosols and the radiative forcing induced [Graf et al. (1997, 1998), Gasso (2008), Ge et al. (2016)]. In contrast, few studies focus on the impact on tropospheric composition including air quality, with the exception of case studies of volcanic eruptions [e.g. Colette
et al. (2010), Schmidt et al. (2015), Boichu et al. (2016, 2019)].

In this context, the objective of this work will be focused on the study of the impact of volcanic sulfur emission on the tropospheric composition and on surface species concentration and deposition, at the global scale. We want to assess and analyse the contribution of volcanoes to the global sulfur budget using a Chemistry-Transport Model (CTM). Here, we use the MOCAGE CTM, developed at the Centre National de Recherches Météorologiques (CNRM) [Josse et al. (2004),Guth (2015)].
Firstly, we will evaluate the changes induced by the update of the volcanic sulfur emission inventory into MOCAGE; precisely from the inventory of Andres and Kasgnoc (1998) to the one from Carn et al. (2016, 2017). Secondly, the focus will be on the analysis of the volcanic $SO_2$ and sulfate aerosol tropospheric distribution and contribution, at the global scale, as well as the sulfur species concentration and deposition at the surface.

In Section 2, we present the simulations configuration of MOCAGE CTM. The new volcanic $SO_2$ emission inventory and
its upgrades compared to the Andres and Kasgnoc (1998) one are described in Sect. 3. In Sect. 4 are presented the setup of the simulations and the observations used to evaluate them. The evaluation of the updating inventory and the comparison in tropospheric and surface species concentrations between the simulations are presented in Sect. 5. Then, the new sulfur species distribution and budget in the atmosphere are shown in Sect. 6. A sensitivity analysis on the passive emission sources based on the annual uncertainties provided is carried out in Sect. 7. Finally in Sect. 8, a conclusion is given.

## 2    Description of MOCAGE model

### 2.1    General features

MOCAGE (Modèle de Chimie Atmosphérique à Grande Échelle) is an off-line global and regional three-dimensional chemistry-transport model developed at CNRM [Josse et al. (2004), Guth (2015)]. Its use is applied to various scientific topics: impact of climate change on atmospheric composition [e.g. Teyssèdre et al. (2007), Lacressonnière et al. (2014, 2016, 2017), Lamarque
et al. (2013)], chemical exchanges between the stratosphere and the troposphere using data assimilation [e.g. El Amraoui et al. (2010), Barré et al. (2012)], operational use for air quality forecasting for France (Prev'Air program [Rouil et al. (2009)]) and for Europe (as one of the nine models contributing to the regional ensemble forecasting system of the Copernicus Atmosphere Monitoring Service (CAMS) european project [Marécal et al. (2015); https://atmosphere.copernicus.eu/]).

A special feature of the model makes it possible to include a natural or anthropogenic accidental source, such as volcanic
eruptions or nuclear explosions, during a simulation. This feature is used as part of the Toulouse VAAC (Volcanic Ash Advisory Center) of Météo-France, which is responsible for monitoring volcanic eruptions over a large area (including part of Europe




and Africa). In order to input an accidental emission, the time and place (latitude/longitude), the bottom and top plume heights, the total quantity emitted as well as the duration emission are required.

## 2.2 Model geometry and inputs

The CTM MOCAGE can be used with global or regional resolutions based on its grid nesting capability. Each outer domain forces the inner domain at its edges (boundary conditions). The global domain has a typical resolution of $1°$ longitude x $1°$ latitude (around $110\,\mathrm{km}$ x 110 km at the equator and 110 km x 80 km at mid-latitudes), while the regional domain resolutions are typically $0.2°$ longitude x $0.2°$ latitude (around 22 km x 16 km at mid-latitudes) and $0.1°$ longitude x $0.1°$ latitude resolution (around 11 km x 8 km at mid-latitudes).

The vertical grid has 47 levels from the surface to 5 hPa (about 35 km), with 7 levels in the planetary boundary layer, 20 in the free troposphere and 20 in the stratosphere. The vertical coordinates are expressed in $\sigma$-pressure; meaning that the model levels follow closely the topography in the low atmosphere and the pressure levels in the upper atmosphere.

Being an off-line model, MOCAGE gets its meteorological fields (wind speed and direction, temperature, humidity, pressure, rain, snow and clouds) from an independent numerical weather prediction model. In practice, they can come from two

meteorological models: the IFS model (Integrated Forecast System) operated at the ECMWF (European Center for Medium-Range Weather Forecast System, http://www.ecmwf.int) or from ARPEGE model (Action de Recherche Petite Echelle Grande Echelle) operated at Météo-France [Courtier et al. (1991)].

## 2.3 Emissions

At the global scale, anthropogenic emissions from MACCity inventory are used [Lamarque et al. (2010)], while biogenic emis-

sions for gaseous species are from MEGAN-MACC inventory, also representative of the year 2010 [Sindelarova et al. (2014)]. Nitrous oxides from lighning are based on Price et al. (1997) and parameterized dynamically according to the meteorological forcing. Organic and black carbon are taken into account following MACCity [Lamarque et al. (2010)]. DMS oceanic emissions are a monthly climatology, $1°$horizontal data [Kettle et al. (1999)]. Finally, biomass burning are from the daily GFAS products [Kaiser et al. (2012)].

In MOCAGE, with the exception of the species from biomass burning process [Cussac et al. (2020)], lightning $NO_x$ [Price et al. (1997)] and aircraft emissions [Lamarque et al. (2010)], all of the chemical species sources are injected, for numerical reasons, in the first five levels of the model. The injection profile of anthropogenic and biogenic emissions follows an exponential decrease from the surface level of the model: $\delta_L = 0.5\delta_{L+1}$, with $\delta_L$ the injection fraction of the mass emitted at the level L of the model; meaning that the majority of pollutants are emitted at the surface and then quickly decrease in altitude.



## 2.4 Chemistry and aerosols

### 2.4.1 Gaseous species

The MOCAGE chemical scheme uses a compilation of two chemical schemes in order to represent both the tropospheric and stratospheric air composition. The first one, the Regional Atmospheric Chemistry Mechanism (RACM) [Stockwell et al. (1997)] with the sulfur cycle completed [details in Guth et al. (2016)], is used in the troposphere. The second one, REactive Processes Ruling the Ozone BUdget in the Stratosphere (REPROBUS), is used in the stratosphere [Lefèvre et al. (1994)].

A total of 112 gaseous compounds, 379 thermal gaseous reactions and 57 photolysis rates are represented in MOCAGE. The calculation of the reaction rates is performed during the simulation every 15 min. The photolysis reaction rates are interpolated on the same 15-minute time step from a look-up table precalculated from TUV model [Madronich (1987)]. A modulation at each grid point and for all time iterations is applied as a function of the ozone column, solar zenith angle, cloud cover and surface albedo.

### 2.4.2 Aerosols

Both primary and secondary aerosols are represented in the model [Martet et al. (2009), Sič et al. (2015), Guth et al. (2016), Descheemaecker et al. (2019)]. All types of aerosols use the same set of six sectional size bins, ranging from 2 nm to 50 $\mu$m (with size bins limits of 2, 10, and 100 nm, and 1, 2.5, 10 and 50 $\mu$m).

Primary aerosols are composed of four species: black carbon, primary organic carbon, sea salt and desert dust. The first two species (black and organic carbon) depend on emission inventories while sea salts and desert dusts are dynamically emitted using the meteorological forcing at the resolution of each domain [Sič et al. (2015)].

Secondary inorganic aerosols (SIA) are implemented in MOCAGE [Guth et al. (2016)]: sulfate, nitrate and ammonium aerosols. The thermodynamic equilibrium model ISORROPIA (precisely the lastest version ISORROPIA II) [Nenes et al. (1998), Fountoukis and Nenes (2007)] is used to calculate SIA concentrations in MOCAGE depending on the partition of compound concentrations, the gaseous and aerosol phases, and the ambient conditions (temperature, pressure).

Secondary organic aerosols (SOA) are estimated from the partition between particulate organic compounds and black carbon measured into observation samples [Castro et al. (1999)]. Therefore it is implemented in the model as a ratio of the primary carbon species in the emission input [Descheemaecker et al. (2019)].

## 2.5 Transport

The transport in the model is solved in two steps. A first one explicitly determines the large-scale transport (advection) with the wind input data provided by the forcing weather model. For this purpose, a semi-Lagrangian scheme is used [Williamson and Rasch (1989)]. The second step parameterizes subgrid phenomena that cannot be solved explicitly, such as convection and turbulent scattering. The convective transport is parameterized upon Bechtold et al. (2001) parameterization. The scheme of Louis (1979) is used to diffuse the species by turbulent mixing.





## 3 Volcanic sulfur emissions in the model

Volcanic emissions are composed of several gases, with the chemical composition changing from a volcano to another depending on the geodynamical context. Among them, sulfur species emitted by volcanoes are mainly sulfur dioxide ($SO_2$) and hydrosulphuric acid ($H_2S$) in much lower quantity. Being by far the dominant species, only $SO_2$ is referenced in global
inventories of volcanic emissions.

### 3.1 Current volcanic sulfur inventory

The current inventory implemented in MOCAGE is from Andres and Kasgnoc (1998), a study contributing to the work of GEIA. It was carried out over a period of about 25 years, from the early 1970s to 1997, and based on measurements of volcanic $SO_2$ emissions at the global scale.
A synergy between the COSPEC surface instrument and the TOMS satellite instrument was used. The COSPEC (COrrelation SPECtrometer) is a correlation spectrometer initially used in pollution measurements [Moffat and Millan (1971), Williams-Jones et al. (2008)]. However, volcanologists have adapted it to measure the quantities of sulfur dioxide in a moving air mass (here the volcanic plume). It works by comparing the amount of solar ultraviolet (UV) radiation absorbed in the plume with a standard (one sample of the background sky and two laboratory-calibrated $SO_2$ concentration cells). It is most commonly used
under calm to moderate eruptive conditions. On the contrary, the space instrument TOMS (Total Ozone Mapping Spectrometer) [Krueger et al. (1995), Seftor et al. (1997), Torres et al. (1998a)], operational between 1978 and 2005, was capable to detect larger eruptions thanks to the similar molecular structure of $SO_2$ and ozone ($O_3$). Thus, during volcanic eruptions emitting large amounts of sulfur dioxide, a quantitative $SO_2$ measurement can be extracted from an "apparently" measured ozone signal in the UV band from 300 to 340 nm [details of the algorithm in Krueger et al. (1995)]. The synergy of these two
instruments is therefore complementary in the development of the inventory. Although the first instrument is better adapted to the measurement of weak flares and the second to the strongest ones, a campaign dedicated to Popocatepetl in Mexico showed the good correlation between the two instruments [Schaefer et al. (1997)].

Measurements were only carried out on subaerial volcanoes, i.e. emitting gases directly into the atmosphere. A total of 69 volcanoes are listed in the inventory, divided into two categories: 49 continuously erupting volcanoes and 25 sporadically
erupting volcanoes. Five volcanoes belong to both categories because they had a main activity of continuous emission but punctuated by eruptive events: Aso, Augustine, Kilauea East Rift Zona, Mayon and San Cristobal.

Since the beginning of volcanic emission measurements in the early 1970s, the global activity of continuous eruptions has shown relative constancy. Thus, the fluxes provided in the inventory, in order to incorporate natural variations due to temporal and even chemical inhomogeneities, correspond to a temporal average of all measurements for each volcano. Only three
volcanoes are not concerned by this hypothesis: Etna in Sicily, Kilauea and the Kilauea Rift Zone in Hawaii, which are known as the one of the largest passive emitters of $SO_2$. For them, a flux provided by specific studies [personnaly communicated to Andres and Kasgnoc (1998)] supersedes this average.





With regard to sporadic eruptions, Andres and Kasgnoc (1998) explain that a marked increase is seen in the activity during the dataset time period. Therefore, only the maximum measurements recorded during the period are included.

Knowing that sporadic eruption data in Andres and Kasgnoc (1998) inventory are not recent, it is not possible to take them into account for the recent year chosen for the MOCAGE simulation. Therefore, only continuous eruptions are used in MOCAGE and a global time-averaged $SO_2$ flux of 13 Tg.yr$^{-1}$ is reported.

No information is available on plume heights. Thus, as for the other chemical species emitted, volcanic $SO_2$ is emitted at the lowest levels of the model (see Sect. 2.3).

**3.2    New volcanic sulfur inventory**

With the technological improvements in satellite technology, an increasing number of satellites are now able to better detect the sources of volcanic $SO_2$: plume heights, quantities emitted and location. The most recent instruments [e.g Ozone Monitoring Instrument (OMI)] have a higher sensitivity to detect much smaller and less emitting sources, but also passive degassing, with respect to TOMS. Global coverage gives another considerable advantage over other measurement techniques. As a reminder,
COSPEC carries out measurements from the ground. It cannot therefore cover all of the Earth's volcanoes, which are sometimes hard to access. The use of satellites thus makes it possible to measure the $SO_2$ emission fluxes of a larger number of volcanoes.

Carn et al. (2016, 2017) work update and complete Andres and Kasgnoc (1998) work with a new inventory. The inventory is divided into two parts corresponding to the two types of emissions detectable by satellites: eruptive emissions (given as daily total $SO_2$ emitted) and passive degassing (given as annual mean emissions).

First, the eruptive emissions dataset [Carn et al. (2016) data available in Carn (2019)] is a synthesis of 40 years of daily $SO_2$ measurements (between 31/10/1978 and 31/12/2018) derived from 7 satellite instruments in the ultraviolet (UV) and infrared (IR) range. Data from 119 volcanoes and a total of 1502 events over the period are provided. For each of these eruptions, the information given is the location of the volcano (latitude and longitude), the date and time, the VEI (Volcanic Explosivity Index), the measured $SO_2$ mass released (in kt), but also the height of the volcano and the height of the plume (measured if
possible, estimated if not). We will use the additional information from Carn et al. (2016) on the injection height, taking into account the height of the volcano as the base of the emissions and the height of the plume as the top of the injection. Also, the daily frequency allows us to take into account the eruptions in simulations for the period covered by the inventory.

Secondly, the passive degassing dataset is the first documented volcanic sulfur dioxide emission inventory made with global satellite measurements [Carn et al. (2017)]. It was retrieved from the observations of OMI instrument in the UV domain during
a long-term mission between 2005 and 2015. The high sensitivity of the instrument was a technological breakthrough that could distinguish low $SO_2$ sources; $\sim$30 kt/y for persistent anthropogenic sources and lower ($\sim$6 kt/y) for volcanoes which are located at higher altitudes or at lower latitudes that benefit from more satellite observations and optimal conditions (low solar zenithal angle). The volcanic $SO_2$ sources have been identified on the basis of 3-year averages (2005-2007, 2008-2010, 2011-2014), which implies that for a source to be characterized as persistently degassing, the emission must be relatively constant
on this time scale. Then, annual mean emissions were calculated for each source over the 11 years of the study (from 2005 to 2015 included).



Moreover, uncertainties over annual average fluxes are available, resulting from the measurement process, the calculation algorithm and from the characterization of the type of emission. The total uncertainty of annual sulfur dioxide fluxes are estimated at 55 % and over 67 % for sources emitting more than 100 kt/y and less than 50 kt/y, respectively. This latter
information will be exploited in the sensitivity analysis (see Sect. 7).

The dataset provides annual averages per year of the $SO_2$ flux over about 90 volcanoes. We assume that emission fluxes are constant every day of the year.

A major improvement of the Carn et al. (2016, 2017) volcanic emission inventory is the availability of the plume height estimation in case of an eruption, as well as the altitudes of the different volcanoes referenced. We implemented a parameter-
ization of the injection heights according to the type of emissions, meaning that it is no longer emitted at the model surface as for the current volcanic emissions. Each eruption is emitted from the model level of the volcano altitude to the level of the plume top height, while passive degassing will be emitted only at the model level of the volcano altitude. This is an important improvement, knowing that in some areas, depending on the model resolution chosen, the model orography may differ from the actual topography. This can have an impact on the transport of volcanic emissions.

The vertical distribution of the emission mass has to be set in the model. A study by Stuefer et al. (2013) proposes a realistic representation (for ash and $SO_2$ plumes) that we implemented in MOCAGE. During a volcanic eruption, the emitted materials are rapidly transported vertically by the convection in the plume, since the emitted gases are warmer than the surrounding atmosphere. Thus, most of the materials is concentrated at high altitude, giving the plume an "umbrella" profile. In practice, the plume follows an almost linear profile from the top of the crater and then opens into a parabola containing 75 % of the
gases in mass into the top third of the plume.

In summary (see Table 1), the update of the volcanic sulfur emission inventory compiles now about 160 volcanoes (∼110 in the eruptive category and ∼90 in the passive degassing category with 40 volcanoes in common), spread over the globe, which emit quantities of $SO_2$ detectable by satellites. Finally, the availability of emission heights in this inventory gives a better description of the emission.

**4 Simulation set-ups and observations**

**4.1 Description of the simulations**

Four different simulations are carried out in order to evaluate the impact induced by the update of the volcanic $SO_2$ inventory in MOCAGE and its contribution to the sulfur species budget in the atmosphere at the global scale. The four simulations are run at a resolution of 1° lon x 1° lat.

Meterological fields are driven by the ARPEGE 3-hourly forecasts. For all simulations, the same emission inventories are used, with the exception of the volcanic one. The same global annual sulfur emissions are computed for all other sources. Anthropogenic and biomass burning sources emit $SO_2$, whereas biogenic emissions from the ocean are assumed to occur as DMS. Oceanic DMS emissions are 19.93 Tg S.yr[-1], while anthropogenic emissions are 48.64 Tg S.yr[-1]. For 2013, biomass burning emissions from GFAS products were relatively low, only 0.99 Tg S.yr[-1].





**Table 1.** *Summary information on the current [Andres and Kasgnoc (1998)] and the updated [Carn et al. (2016, 2017)] SO₂ volcanic emission inventories.*

|  | Current Volcanic Inventory | New Volcanic Inventory | |
|---|---|---|---|
|  | Andres and Kasgnoc (1998) | Carn et al. (2016) | Carn et al. (2017) |
| Emission type | Continuous emissions | Eruption | Passive degassing |
| Period | 1970-1997 | 1978-2018 | 2005-2018 |
| Instruments | COSPEC & TOMS | TOMS & OMI | OMI |
| Frenquency | Time averaged over the period | Daily total quantity per volcano | Annual mean quantity per volcano |
| Information on the vertical | no information | volcano altitude | volcano altitude & plume height |
| Nb of volcano | 43 | 119 | 91 |

**Table 2.** *Characteristics of the simulations.*

|  | Volcanic inventory | Altitude of injection |
|---|---|---|
| NOVOLC | N/A | N/A |
| REF | Andres and Kasgnoc (1998) | at model surface |
| CARN | Carn et al. (2016, 2017) | at model surface |
| CARNALTI | Carn et al. (2016) - eruption | from volcano vent to plume top |
|  | Carn et al. (2017)- degassing | at volcano vent |

250  Concerning volcanic sulfur emission inventories, either Andres and Kasgnoc (1998) inventory or Carn et al. (2016, 2017) inventories are used. The full eruption emission database is available following Carn (2019) (https://disc.gsfc.nasa.gov/datasets/ MSVOLSO2L4_3/summary).

  The characteristics of the four simulations are given in Table 2. One simulation takes into account only anthropogenic emissions (no volcanic emission), named NOVOLC. Another one is run with anthropogenic emissions and the current volcanic
255 inventory [from Andres and Kasgnoc (1998)], named REF. The two last simulations take into account anthropogenic emissions and the new volcanic inventory [from Carn et al. (2016, 2017)]. However, one injects the volcanic SO₂ emissions at the surface, named CARN as currently done in MOCAGE, and the other one uses the new parameterization to inject the emission in altitude (from the volcano altitude up to the plume height with the "umbrella" profile), named CARNALTI.

  Three simulations will be used to compare and evaluate the effect induced by the update of the volcanic sulfur inventory
260 in MOCAGE. We will compare REF, CARN and CARNALTI experiments to highlight both the changes brought by the new inventory and the parameterization to inject the volcanic emissions in altitude.

  Then, we will analyse more deeply two simulations, CARNALTI and NOVOLC, in order to study the sulfur budget in the atmosphere modeled by MOCAGE. CARNALTI simulation being the most realistic one by construction, it should provide





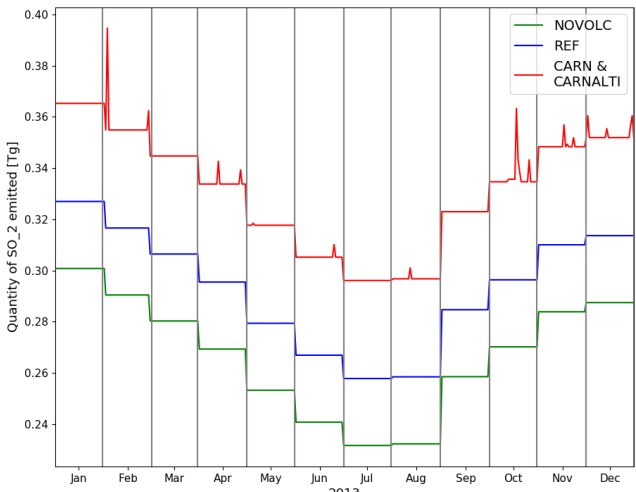

**Figure 1.** *Temporal evolution of 2013 annual SO₂ emissions in Tg; anthropogenic inventory for* NOVOLC, *plus Andres and Kasgnoc (1998) volcanic inventory in* REF *or Carn et al. (2016, 2017) volcanic inventory in* CARN *and* CARNALTI.

the best representation of the global tropospheric composition. And in order to look specifically at the contribution of the

volcanoes in the sulfur budget of the atmosphere, we will compare CARNALTI to the simulation with no volcanic sulfur emissions (NOVOLC).

The four simulations are run for the year 2013 with a 3 month spin-up period (from october to december 2012). In addition to being one of the years when a large amount of observational data is available globally, 2013 is chosen as the year with the lowest eruptive emission flux in Carn et al. (2016) inventory. Therefore, the eruptive emission are negligible in 2013. This

makes the comparison between REF and CARN AND CARNALTI more relevant, since no eruptions are taken into account in REF. Figure 1 shows the annual volcanic emissions from the different simulations. We notice the monthly variation due to the anthropogenic emissions in NOVOLC simulation, with less emissions during the summer and the highest values in winter. The current inventory [Andres and Kasgnoc (1998)] implemented in REF simulation adds constant volcanic emissions to the anthropogenic ones. However, we can see in CARN and CARNALTI simulations, a strong increase in the volcanic emissions

due to the passive degassing as well as the daily variation due to the sporadic eruptions referenced. Indeed, Andres and Kasgnoc (1998) inventory counts 13 Tg of $SO_2$ emissions per year (or 6.5 Tg S), while the total 2013 annual emissions into Carn et al. (2016, 2017) inventory are 23.68 Tg of $SO_2$ (or 11.84 Tg S), with 23.48 Tg of passive degassing $SO_2$ and 0.20 Tg of eruptive emission (< 1 % of the total amount of volcanic $SO_2$ emission).

Figures 2 represents spatially the difference between the current and the new inventory. The red dots mostly show new

volcanoes in Carn et al. (2016, 2017) inventory against Andres and Kasgnoc (1998) one. However, we also notice blue dots, meaning that in the new inventory, the estimated emission fluxes are reduced. Given the low number of eruptive emissions in 2013, the annual average of volcanic emissions in Fig. 2 essentially represents passive emissions.





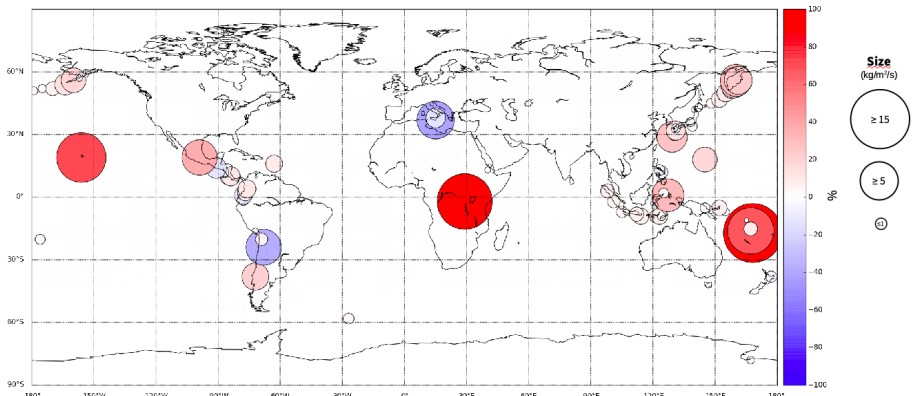

**Figure 2.** *2013 annually average ratio between volcanic $SO_2$ emissions in Carn et al. (2016, 2017) and Andres and Kasgnoc (1998) inventories. The size of the rounds represents the absolute difference while the color represents the relative difference.*

## 4.2 Observations used for the evaluation of the simulations

Due to the observational data available regarding sulfur species for the evaluation of the simulations, the target species will
be $SO_2$ and aerosols, since $SO_2$ is the precursor of sulfate aerosols. Space-borne observations have the benefits to provide a
global coverage. Several satellite-based instruments can be used to validate the simulations, but only intruments different from
those used to set the inventories can be selected for an independent evaluation. Based on this constraint, we use GOME-2
for $SO_2$ total columns. For aerosols, we use MODIS (Moderate-Resolution Imaging Spectroradiometer) aerosol optical depth,
indirectly correlated to $SO_2$.

### 4.2.1   $SO_2$ total column GOME-2

Global GOME-2 daily mean total $SO_2$ column from GOME-2/Metop A Level 2 were used to evaluate the model simulations [ACSAF (2017)]. This product is available at the global scale from January 2007 to November 2016. The retrieval of
the $SO_2$ product is based on irradiance measurements, between 315-326 nm, from a DOAS (Differential Optical Absorption
Spectroscopy) technique. The DOAS algorithm is used to determine the trace gas slant column and produce vertical columns
thanks to fitting AMF (Air Mass factor). Various physical or technical causes can reduce the quality of the total column $SO_2$
measurement of the GOME-2 product. Thus, following the recommendations of Valks (2016), a filter is applied to remove measurements in polar regions (Solar Zenithal Angle (SZA) > 75°), measurements in the South Atlantic Anomaly region [Richter
et al. (2006)], pixels with a cloud fraction greater than 30 % and data unavailable due to unfulfilled external data dependency.

Plus, another well known presence of offsets in the data lead us to add another selection criteria based on high pass filtering
along individual orbits [Krotkov et al. (2006)]. This approach consists in eliminating the background noise by isolating the
"real" high $SO_2$ concentrations from the measurements for each orbit (notably due to real volcanic or anthropogenic signals).
From the remaining signal, the median value per latitude range is computed and then subtracted at each measurement of the





orbit, with the exception of the "real" high concentrations previously isolated. This allows us to smooth the background values by constraining them to be close to zero, while keeping the true information of the high $SO_2$ concentrations. However, the data

are still relatively noisy, especially in the midlatitudes and on intersecting orbit trajectories. Figures S1a and S1b highlight the changes in GOME-2 data before and after this background treatment.

### 4.2.2 Aerosol Optical Depth MODIS

The global MODIS daily AODs (*Aerosol Optical Depth*) from MODIS data level 3 (L3, collection 6) were used to evaluate the model simulations. Additionally, a quality control and screening based on Sič et al. (2015) is performed in order to consider

only the best possible observations. It means that to avoid observations with artificially increased AOD, cloud contaminated and statistically low confident observations are removed [Zhang et al. (2005), Koren et al. (2007), Remer et al. (2008)]. Moreover, all AOD values below 0.05 are automatically filtered because Ruiz-Arias et al. (2013) highlighted the rapid growth in the relative underestimation of AODs after this threshold which leads to a mean relative error above 50 %.

In MOCAGE, AODs are calculated using Mie theory with the Global Aerosol Data Set's refractive indices [Köpke et al.

(1997)] and extinction efficiencies derived with the Mie scattering code for homogeneous spherical particles from Wiscombe (1980).

### 4.3 Statistical metrics used for evaluation

In order to evaluate the model against observation data, we can use several statistical metrics. We chose the fractional bias, the fractional gross error and the correlation coefficient, following Seigneur et al. (2000). Indeed, using the bias and the root-

mean-square error (RMSE) can provide misleading information when the denominator is small compared to the numerator.

The fractional bias or modified normalized mean bias (MNMB) quantifies the mean between the modelled ($f$) and the observed ($o$) elements, for $N$ observations. It ranges between -2 and 2 and varies symetrically with respect to under and overestimation of the model. The definition is given by:

$$MNMB = \frac{2}{N} \sum_{i=1}^{N} \frac{f_i - o_i}{f_i + o_i} \tag{1}$$

The fractional gross error (FGE) quantifies the model error. It is a positive variable ranging between 0 and 2. The definition is given by:

$$FGE = \frac{2}{N} \sum_{i=1}^{N} \mid \frac{f_i - o_i}{f_i + o_i} \mid \tag{2}$$





The correlation coefficient ($r$) indicates whether the variations of the model and the observations are well matched and ranges between -1 and 1. The closer the score is to 0, the weaker the correlation is. The definition is given by:

$$r = \frac{\frac{1}{N}\sum_{i=1}^{N}(f_i - \overline{f})(o_i - \overline{o})}{\sigma_f \sigma_o} \tag{3}$$

where $\overline{f}$ and $\overline{o}$ are, respectively, the model and observations mean values, and $\sigma_f$ and $\sigma_o$ are the standard deviations from the modelled and observed time series.

## 5 Evaluation and impact of the volcanic emission inventory update

### 5.1 Evaluation of the simulations

The simulation evaluation against space-borne observations is done both at the global and local scales. Even if the evaluation at the global scale enables to see the overall changes in the model, we expect noticeable changes at the local scale, in the vicinity of the volcanoes.

Therefore, three zones are chosen to complete the evaluation at the global scale. These zones are chosen as areas with different types of changes between Andres and Kasgnoc (1998) and Carn et al. (2016, 2017) volcanic emissions inventories. Plus, these areas reference few, if only one volcano, per zone which are among the largest passive $SO_2$ emitters in Carn et al. (2017) inventory.

The Zone 1 is centered over Central Africa and under the influence of the Nyiragongo and Nyamuragira (alt. 2950 m). In Andres and Kasgnoc (1998) inventory, this volcano is not listed. On the contrary, in Carn et al. (2017), the passive degassing emission represents 2.29 Tg in 2013. No eruption is listed in Carn et al. (2016) inventory for 2013.

The Zone 2 is located in the North Pacific Ocean, around Hawaï. The volcano, based on the island, is the Kilauea (alt. 1222 m). In the REF simulation, the volcano emissions in the inventory are 0.45 Tg.yr$^{-1}$ (7th rank of the most $SO_2$-emitting volcanoes in Andres and Kasgnoc (1998) inventory). But in Carn et al. (2017), the Kilauea emissions are updated and it is the second biggest emitter with 2.17 Tg. In 2013, no eruption are recorded in Carn et al. (2016) inventory for this area.

The Zone 3 is located in the Mediterranean region, under the influence of the Etna (alt. 2711 m) and Stromboli (alt. 870 m). In Andres and Kasgnoc (1998) inventory, 1.48 Tg.yr$^{-1}$ are emitted by the Etna (the biggest volcanic $SO_2$ emitter referenced), 0.27 Tg.yr$^{-1}$ are emitted by the Stromboli and also 0.02 Tg.yr$^{-1}$ by the Vulcano. In Carn et al. (2016, 2017) inventory, only 0.65 Tg of $SO_2$ are emitted in 2013 in the Zone 3; corresponding to less than 0.04 Tg for Stromboli and 0.61 Tg for the Etna. The Vulcano is not inventored in Carn et al. (2016, 2017) inventory, at all. In 2013, small eruptions have occured at Etna, counting a little less than 0.06 Tg. Therefore, in the updated Carn et al. (2016, 2017) inventory, volcanic emissions in Zone 3 are weaker than in Andres and Kasgnoc (1998) inventory.





**Table 3.** *SO$_2$ total column statistics of* REF*, CARN and* CARNALTI *simulations against GOME-2 observations at the global scale.*

|  | MNMB | FGE | Correlation ($r$) |
|---|---|---|---|
| REF | -0.822 | 1.131 | -0.166 |
| CARN | -0.792 | 1.118 | -0.170 |
| CARNALTI | -0.743 | 1.078 | -0.165 |

**Table 4.** *SO$_2$ total column statistics of* REF*, CARN and* CARNALTI *simulations against GOME-2 observations at the specifics zones.*

|  | Zone 1 | | | Zone 2 | | | Zone 3 | | |
|---|---|---|---|---|---|---|---|---|---|
|  | MNMB | FGE | Coorelation ($r$) | MNMB | FGE | Correlation ($r$) | MNMB | FGE | Correlation ($r$) |
| REF | -0.75 | 0.95 | -0.29 | -0.38 | 0.98 | 0.09 | 0.74 | 0.86 | 0.51 |
| CARN | -0.27 | 0.59 | 0.21 | 0.62 | 0.94 | 0.39 | 0.55 | 0.74 | 0.40 |
| CARNALTI | -0.26 | 0.60 | 0.75 | 1.03 | 1.08 | 0.48 | 0.56 | 0.73 | 0.37 |

### 5.1.1 Validation against GOME-2 SO$_2$ total column

Table 3 presents the statistics against GOME-2 SO$_2$ total column at both the global scale. The MNMB is better in CARNALTI simulation with a value of -0.743, higher than -0.822 and -0.792 in REF and CARN simulations, respectively. Figure 3 represents 2013 annual MNMB at the global scale in REF, CARN and CARNALTI againts GOME-2 SO$_2$ total column (see Fig. S2 for FGE and correlation coefficient). We notice small changes in the vicinity of volcanoes where MNMB score is improved between REF and CARNALTI. Similarly, FGE is better in CARNALTI compared to the other simulations at the global scale. However, the correlation coefficient is almost unchanged between the three simulations.

GOME-2 data observations present strong SO$_2$ concentrations over polluted areas but also at the location of volcanic point sources which match MOCAGE simulations, especially CARNALTI simulation (see Fig. S1). However, anthropogenic pollution over Asia in GOME-2 SO$_2$ total column is shifted westerly with respect to MOCAGE simulations. Thus, MNMB scores in this area are not good (around -2 over West China where GOME-2 measurements over-estimate MOCAGE SO$_2$ tropospheric column and +2 over East China where GOME-2 measurements are under-estimated), likely related to uncertainties on anthropogenic emissions over China.

Table 4 presents the same statistics as in Table 3, but calculated at the three specifics zones. Figures 4 present the annual 2013 MNMB for REF, CARN and CARNALTI experiments against GOME-2 observations at the three zones defined previously (see FGE and correlation coefficient in Figs. S3-S4-S5).

In Zone 1 (left column), the negative MNMB score is clearly related to the lack of information on volcanic emission in REF simulation (MNMB of -0.75 in Table 4). However, in the simulations using Carn et al. (2016, 2017) inventory, we can see the improvement brought by the new inventory. The MNMB increases to -0.27 and -0.26 in CARN and CARNALTI simulations,



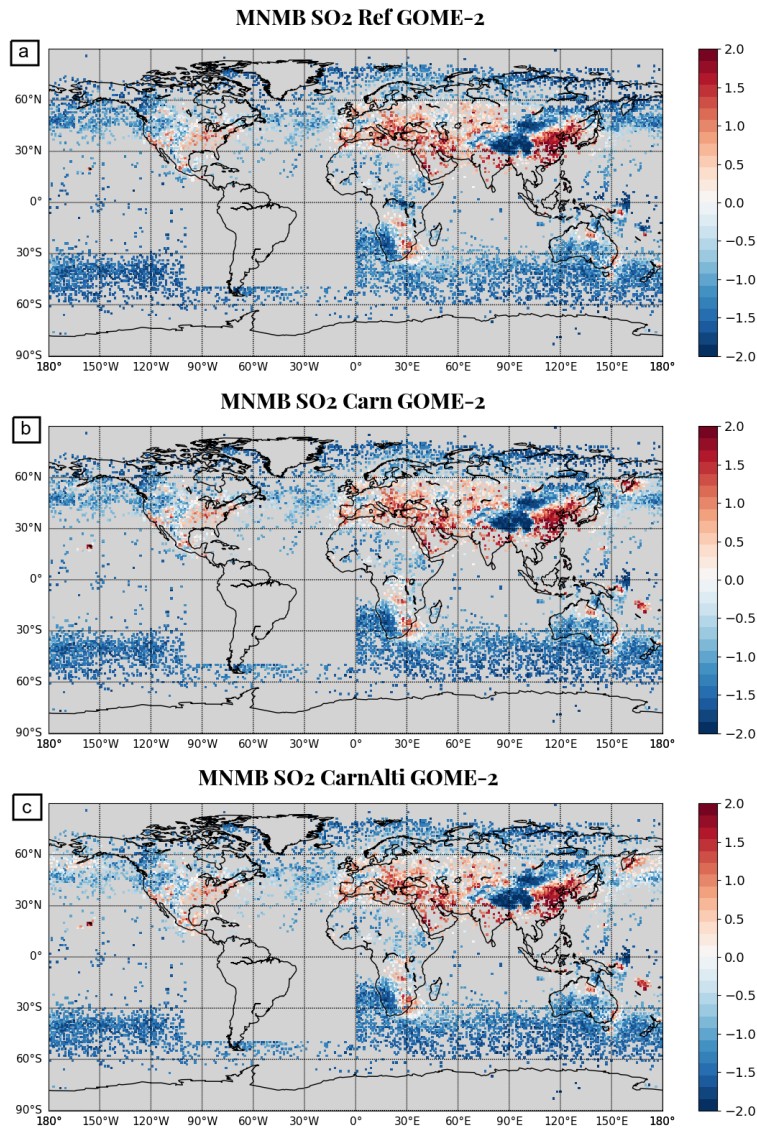

**Figure 3.** *Maps of the 2013 annual MNMB against GOME-2 daily observations for (a)* REF *(b)* CARN *and (c)* CARNALTI *simulations. Grey pixels correspond to the pixels filtered by the processing.*

respectively. The FGE scores are also improved; from 0.95 in REF to 0.60 in CARNALTI. The correlation coefficient is highly improved between -0.29 in REF and 0.75 in CARNALTI simulations.

Similarly for Zone 3 (right column), we note that the decrease in Etna's volcanic emission flux leads to a reduction in the MNMB and FGE in the simulations. For example, the MNMB changes from 0.74 to 0.56 between REF and CARNALTI. However, the correlation coefficient also decreased slightly.



**Figure 4.** *Maps of the 2013 annual MNMB against GOME-2 daily observations for* REF *(top),* CARN *(center) and* CARNALTI *(bottom) simulations for selected zones: (left) Zone 1, (middle (Zone 2) and (right) Zone 3. Grey pixels correspond to the pixels filtered by the processing.*

Zone 2 is the only zone that presents better scores for REF simulation, with the exception of the correlation coefficient. The MNMB is -0.38 and the FGE is 0.98; both better than MNMB of 0.62 and 1.03 and FGE of 0.94 and 1.08 for CARN and CARNALTI, respectively. However, by looking at Fig. 4 (middle column), we can point out that only few pixels of GOME-2 measurements are available in this area. Moreover, the pixels near the volcanic point source already present a positive MNMB score in REF. Thus, with the higher volcanic emission flux implemented in Carn et al. (2016, 2017) inventory, we were
expecting the degradation of the statistical scores for this zone.





**Table 5.** *Year 2013 statistics of* REF*,* CARN *and* CARNALTI *simulations against MODIS observations at the global scale.*

|         | MNMB  | FGE   | Correlation ($r$) |
|---------|-------|-------|-------------------|
| REF     | 0.099 | 0.439 | 0.328             |
| CARN    | 0.110 | 0.436 | 0.331             |
| CARNALTI| 0.149 | 0.425 | 0.345             |

On these 3 specific zones, the MNMB and FGE scores between CARN and CARNALTI are often very close. The correlation coefficient on the contrary shows great improvements between REF, CARN and CARNALTI simulations (*e.g.* in Zone 2, it the the only score showing an improvement). Depending on local conditions, the CARN simulation can therefore sometimes be better than CARNALTI. However, at the global scale CARNALTI gives best results.

**5.1.2 Validation against AOD MODIS**

As a second evaluation step, we compare the simulations with the AOD MODIS. Figure 5 presents for REF, CARN and CARNALTI experiments, the 2013 annual MNMB with respect to MODIS AOD observations. We can see that the equatorial belt has a negative MNMB, between -0.2 and -1.2 in REF simulation, but in CARN and CARNALTI simulations, it is closer to 0; *e.g.* in the vicinity of volcanoes in Indonesia or in central Africa. This shows an improvement of the MOCAGE AOD

modeling at the global scale by updating the volcanic emissions inventory. Despite the improvement in MNMB in the areas near volcanoes, the overall score is not improved (see Table 5). Indeed, the MNMB of the Northern Hemisphere is mainly positive and almost unchanged with the new inventory [Carn et al. (2016, 2017)] where only a few volcanoes are reported. Even this small number of volcanoes, locally, leads to an increase in the already positive MNMB. Thus, globally, the average MNMB is higher in CARNALTI than in REF. Compared to CARNALTI, CARN simulation presents a better MNMB score, due to the

injection of $SO_2$ volcanic emissions near the surface, leading to more important deposition which limits the overestimation in the Northen Hemisphere aerosol concentrations.

Concerning the fractional gross error (FGE), changes are also located in the vicinity of volcanoes (see Fig. S6). In those areas, especially in Central Africa and in Indonesia, the FGE is reduced from a maximum of 1.2 in REF to a maximum of 0.6 in CARNALTI. Globally, the FGE score is improved; 0.439 for REF, 0.436 for CARN and 0.425 in CARNALTI. Even

if, locally in the Northen Hemisphere (*e.g.* in Hawaï), the FGE score can be deteriorated in the simulations with Carn et al. (2016, 2017) inventory, at the global scale, simulations with the new inventory are generally better. CARN simulation against REF only highlights the improvement induced by the global rise of quantity of the $SO_2$ volcanic emission, while the strongest improvement is given by the implementation of the injection in altitude of volcanic emissions.

The correlation coefficient (r) score is better in the Northen Hemisphere (see Fig. S6). Therefore, by adding new volcano

point sources, and mostly in the Southern Hemisphere, the scores are higher for the simulations with Carn et al. (2016, 2017) inventory. The lifetime of aerosols increases when located in higher altitude. Therefore, aerosols are better represented in CARNALTI simulation thanks to the injection in altitude of $SO_2$ (precursor of sulfate aerosols contributing to the AOD).





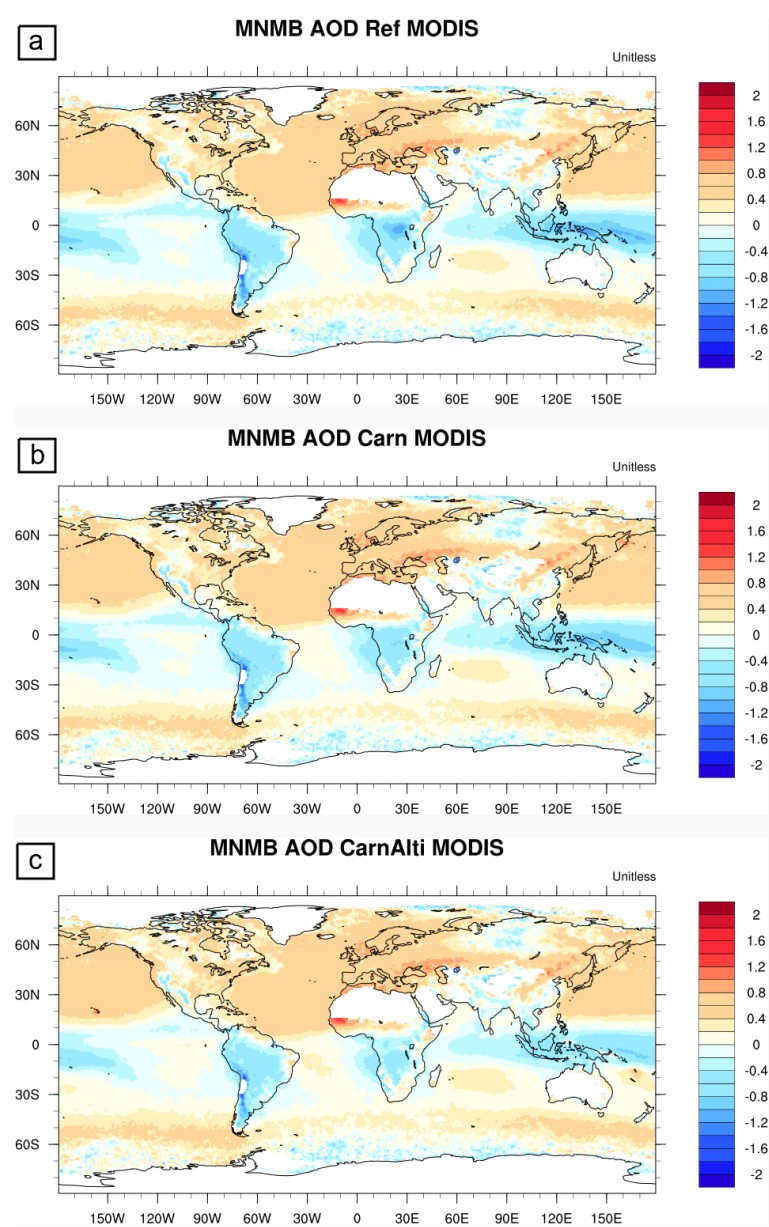

**Figure 5.** *Maps of the 2013 annual MNMB of aerosol optical depth against MODIS monthly observations for (a)* REF *(b)* CARN *and (c)* CARNALTI *simulations.*

Table 6 presents the statistical scores in REF, CARN and CARNALTI simulations against AOD MODIS observations for the three specific zones.

By using Carn et al. (2017) inventory, the model results are improved in Zone 1. Moreover, the statistical scores are even better in CARNALTI simulation using the injection altitude of volcanic emissions. For example, the MNMB raises from -0.53





**Table 6.** *2013 annual statistics of* REF*,* CARN *and* CARNALTI *simulations against MODIS observations on specific zones.*

|  | Zone 1 | | | Zone 2 | | | Zone 3 | | |
|---|---|---|---|---|---|---|---|---|---|
|  | MNMB | FGE | Coorelation ($r$) | MNMB | FGE | Correlation ($r$) | MNMB | FGE | Correlation ($r$) |
| REF | -0.53 | 0.61 | 0.70 | 0.31 | 0.34 | 0.75 | 0.74 | 0.75 | 0.61 |
| CARN | -0.44 | 0.53 | 0.68 | 0.38 | 0.40 | 0.76 | 0.73 | 0.74 | 0.61 |
| CARNALTI | -0.28 | 0.41 | 0.72 | 0.47 | 0.48 | 0.78 | 0.73 | 0.74 | 0.64 |

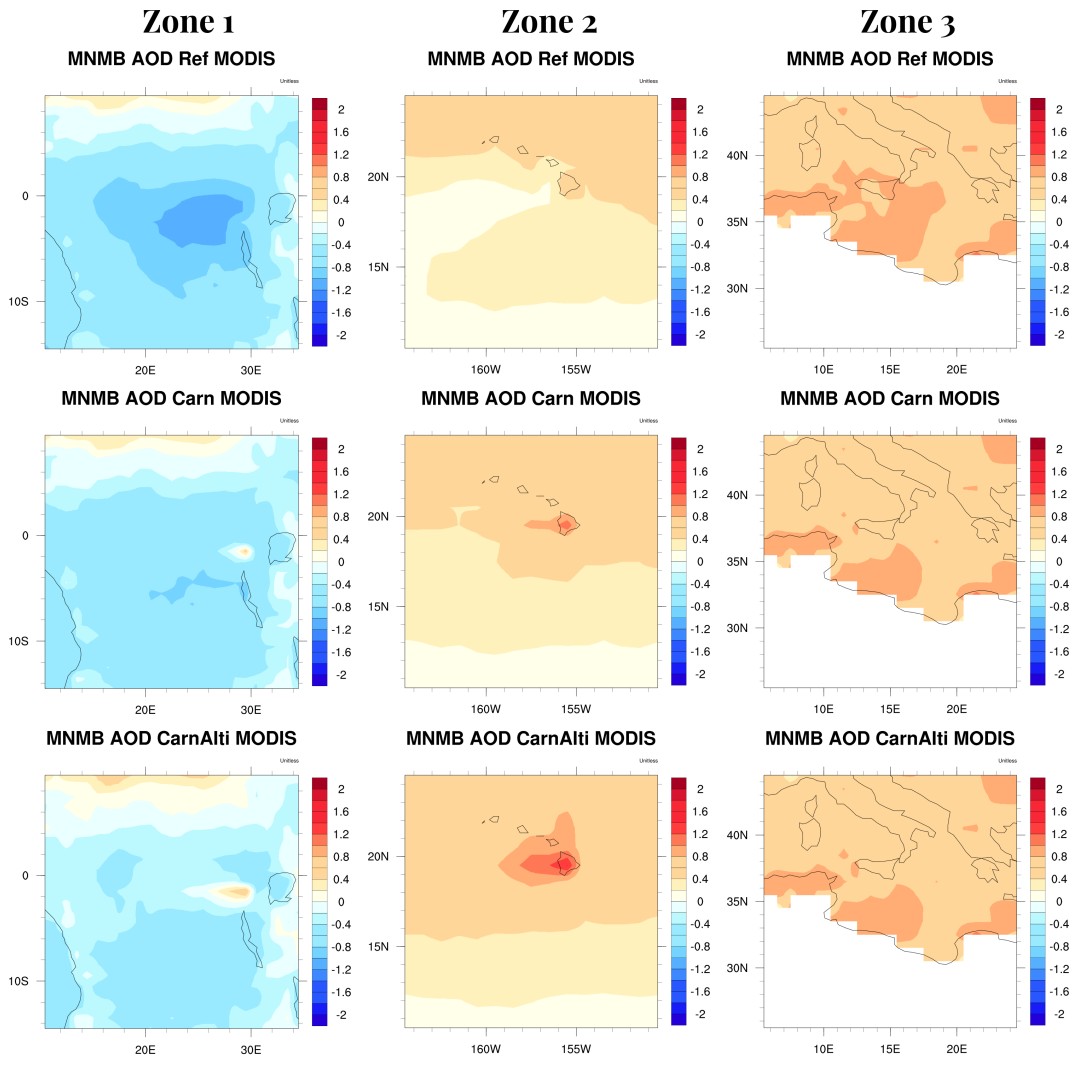

**Figure 6.** *Maps of the 2013 annual MNMB of* REF*,* CARN *and* CARNALTI *simulations against MODIS observations at the specific zones.*





with REF simulation to -0.44 and -0.28 in CARN and CARNALTI simulations, respectively. Similarly, the FGE and correlation coefficient are improved. We notice in Fig. 6 (left column fr Zone 1) that in the REF simulation, a negative MNMB score highlights the lack of the Nyamuragira volcanic $SO_2$ emission. But, with the Carn et al. (2017) inventory, we notice in CARN

and CARNALTI simulations an improvement. The difference brought by the parameterization of injection in altitude, between those two simulations is well seen in the figure. In CARN simulation, the volcanic emission remains close to the volcano. However, the Nyamuragira is at an altitude of around 3000 m, and therefore, its emissions can spread widely in the area as illustrated in CARNALTI.

In Zone 2, unlike the previous area, the MNMB is already positive. Thus, by adding more $SO_2$ volcanic emission, we can see

in Table 6 a deterioration of the MNMB and FGE scores. Only the correlation coefficient increases. With a greater impact on the sulfate aerosols, the CARNALTI simulation presents a more important positive bias. Figure 6 in the middle column confirms those results. However, the volcano being located at an altitude of 1222 m, where the sensitivity of infra-red, mostly, but also ultra-violet instruments is reduce, we can also question the accuracy of the inventory for this volcano, maybe over-estimated.

In Zone 3, the statistical scores are almost similar for all simulations. Indeed, in this region there is a multitude of other

aerosols sources (industries, transport,. . . ). We can see in Fig. 6 a small difference between REF, CARN and CARNALTI simulations. However, the MNMB, FGE and correlation scores are better in CARNALTI, where volcanic gases are emitted just under 3000 m (above the other pollution sources). Thus, using Carn et al. (2016, 2017) inventory and injecting volcanic emissions in altitude improves MOCAGE performances.

## 5.2 Species concentration comparison

$SO_2$, sulfate aerosols and $PM_{2.5}$ tropospheric column and surface concentrations are summarized in Table 7 for each simulations. The annual mean sulfur dioxide total column is 8.27e-6 mol.m$^{-2}$ in CARNALTI simulation, 10 % superior to the 7.51e-6 mol.m$^{-2}$ in REF. Regarding aerosols species, sulfate total column is 23 % higher in CARNALTI simulation but only 2 % for $PM_{2.5}$, because only partially composed of sulfate. This increase is explained by the greater amount of $SO_2$ emitted in Carn et al. (2016, 2017) inventory and by the new injection parameterization. At higher altitudes, the lifetime of sulfur species

is longer due to slower removal processes [Stevenson et al. (2003)]. Figure 7a illustrates it well. It shows the relative difference of sulfate tropospheric column between CARNALTI and REF experiments. We clearly see an increase in CARNALTI concentrations in the vicinity of most volcanic point sources.

In comparison, surface concentrations from the simulations show different results. The aerosol species concentrations still increase but less: from 5.65e-10 kg.m$^{-3}$ in REF simulation to 5.88e-10 kg.m$^{-3}$ (5 %) in CARNALTI for sulfate, 1.264e-8 kg.m$^{-3}$

to 1.271e-8 kg.m$^{-3}$ (<1 %) for $PM_{2.5}$ and from 7.18e-9 mol.m$^{-3}$ to 7.23e-9 mol.m$^{-3}$ (<1 %) for $SO_2$.

On the contrary, for $SO_2$, the surface concentration is greater in REF simulation with 7.72e-9 mol.m$^{-3}$; 7 % higher than the 7.22e-9 mol.m$^{-3}$ in CARNALTI simulation. Due to the volcanic emission injected in altitude, we find more $SO_2$ in altitude, where it converts to sulfate and then sediments. This explains why the sulfate quantities are a bit greater at the surface in the CARNALTI simulation than in REF, but almost equal for $SO_2$.





**Table 7.** *Global and local (Zones 1, 2 and 3) 2013 annual mean concentrations in* REF, CARN *and* CARNALTI *simulations. Gases are in mol and aerosols in kg.*

| | | Mean Tropospheric Column | | | Mean Surface Concentration | | |
|---|---|---|---|---|---|---|---|
| | | $SO_2$ | Sulfate | $PM_{2.5}$ | $SO_2$ | Sulfate | $PM_{2.5}$ |
| | | $(mol.m^{-2})$ | $(kg.m^{-2})$ | $(kg.m^{-2})$ | $(mol.m^{-3})$ | $(kg.m^{-3})$ | $(kg.m^{-3})$ |
| Global | REF | 7.51e-6 | 3.25e-6 | 5.75e-5 | 7.18e-9 | 5.65e-10 | 1.264e-8 |
| | CARN | 7.92e-6 | 3.52e-6 | 5.78e-5 | 8.21e-9 | 5.91e-10 | 1.269e-8 |
| | CARNALTI | 8.27e-6 | 4.01e-6 | 5.85e-5 | 7.23e-9 | 5.88e-10 | 1.271e-8 |
| Zone 1 | REF | 7.837e-6 | 3.93e-6 | 5.44e-5 | 4.86e-9 | 5.37e-10 | 6.03e-9 |
| | CARN | 1.283e-5 | 7.06e-6 | 5.88e-5 | 9.06e-9 | 7.51e-10 | 6.35e-9 |
| | CARNALTI | 1.279e-5 | 1.12e-5 | 6.51e-5 | 4.84e-9 | 1.01e-9 | 6.85e-9 |
| Zone 2 | REF | 4.13e-6 | 4.43e-6 | 7,10e-5 | 4.42e-9 | 3.75e-10 | 2.27e-8 |
| | CARN | 1.45e-5 | 8.03e-6 | 7.37e-5 | 1.77e-8 | 1.11e-9 | 2.33e-8 |
| | CARNALTI | 2.25e-5 | 1.31e-5 | 7.92e-5 | 2.74e-9 | 1.21e-9 | 2.34e-8 |
| Zone 3 | REF | 4.74e-5 | 7.66e-6 | 2.443e-4 | 4.39e-8 | 1.67e-9 | 4.75e-8 |
| | CARN | 3.22e-5 | 6.63e-6 | 2.419e-4 | 3.03e-8 | 1.25e-9 | 4.83e-8 |
| | CARNALTI | 3.13e-5 | 6.84e-6 | 2.424e-4 | 2.30e-8 | 1.07e-9 | 4.81e-8 |

As expected, in CARN simulation results, where the volcanic emissions are injected at the first levels of the model, we notice higher concentrations of both sulfur species at the surface. $SO_2$ and sulfate concentrations are 8.21e-9 mol.m$^{-3}$ and 5.91e-10 kg.m$^{-3}$ respectively in CARN simulation, being around 6 % higher for both species compared with REF but 12 % higher, for $SO_2$, against CARNALTI simulation.

     Thus, by injecting volcanic emission in altitude with the new parameterization in the simulation CARNALTI, less sulfur 455 species remain at the surface and therefore aerosols are spread further from the volcanoes (see Fig. 7b, where we can see volcanic plumes 150 to 200 km away from their source location). By updating the volcanic sulfur inventory into MOCAGE, sulfur species concentrations are increased in CARNALTI compared to REF.

     By looking at the local scale, the differences between CARNALTI and REF can be very large. For example, in Zone 2, the $SO_2$ tropospheric column is 5 times larger (+ 445 %) in CARNALTI (from 4.13e-6 in REF to 2.25e-5 mol.m$^{-2}$), 3 times larger 460 (+ 196 %) for the aerosol sulfate total column (from 4.43e-5 to 1.31e-5 kg.m$^{-2}$) and 5 times larger (+ 223 %) for sulfate at the surface (3.75e-10 to 1.21e-9 kg.m$^{-3}$). In Zone 1, changes are also more important compared to the global scale, with 63 % more concentration of $SO_2$ and 185 % higher concentration of sulfate in the atmosphere and 88 % more sulfate at the surface. In Zone 3, there is less impact because it is a more polluted area.

     Concerning particulate matter, the impact of Carn et al. (2016, 2017) inventory at the global scale does not present significant 465 changes, because $PM_{2.5}$ are not composed only by sulfate aerosols but are the sum of all the atmospheric aerosols with diameter

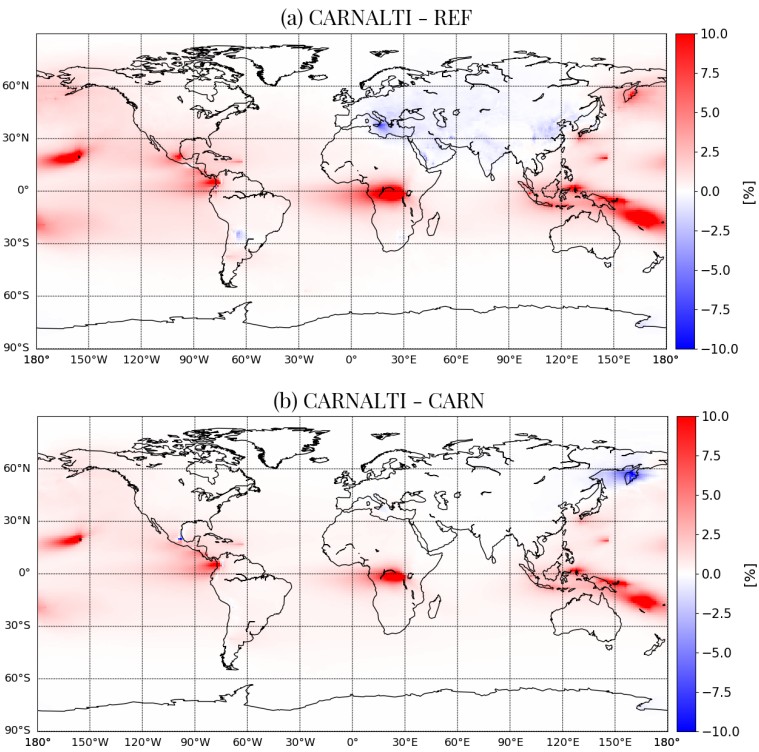

**Figure 7.** *2013 annual mean sulfate tropospheric column relative difference between the (a)* CARNALTI *and* REF *simulations and (b)* CARNALTI *and* CARN *simulations, in %.*

less than 2.5 $\mu$m. However, we can found locally, more important changes; *e.g.* 20 % higher PM$_{2.5}$ tropospheric column concentration in CARNALTI with 6.51e-5 kg.m$^{-2}$ compared to REF with 5.44e-5 kg.m$^{-2}$, in Zone 1. As expected, for Zone 3, all chemical species concentrations are smaller in CARNALTI compared to REF simulation, especially at the surface.

The CARN experiment comparison against CARNALTI presents small differences in the troposheric columns. But the SO$_2$
surface concentration, in all zones, is much higher in CARN simulation compared to CARNALTI due to the injection in altitude of SO$_2$ volcanic emissions (*e.g.* 85 % smaller in Zone 2 in CARNALTI compared to CARN).

## 6 MOCAGE sulfur budget

Considering CARNALTI experiment as the most realistic and best simulation, we retain this one to calculate MOCAGE sulfur budget and analyse the impact of volcanic SO$_2$ emissions on the tropospheric species distribution. In order to isolate the
contribution of volcanic emission from the overall species concentration, we look at the difference between CARNALTI and NOVOLC. More specifically, the contribution of volcanic SO$_2$ emissions to the species budget affected is the quantity of





**Table 8.** *2013 annual global mean SO$_2$ emission, sulfur budget and deposition quantities in Tg. The contribution of sulfur species due to volcanic emissions or other emission sources are presented, in %. The efficiency is the ratio between the contribution of the sulfate burden and the contribution of the total sulfur emission attributed to a specific source. In other words, it is the fractional contribution from anthropogenic and volcanic sources to the sulfate burden.*

| | Sulfur Emission | SO$_2$ Burden | Sulfate Burden | Sulfur Deposition | | | Efficiency |
| --- | --- | --- | --- | --- | --- | --- | --- |
| | | | | Wet | Dry | Sedim | |
| Total (Tg) | 81.41 | 0.17 | 0.82 | 42.41 | 27.81 | 9.80 | - |
| *Sources contributions to the total budget (%)* | | | | | | | |
| Volcanoes | 14.5 | 13.8 | 27.4 | 7.9 | 2.3 | 23.0 | 1.89 |
| Other | 85.5 | 86.2 | 72.6 | 92.1 | 97.7 | 77.0 | 0.85 |

species in the CARNALTI simulation subtracted from the quantity of species in the NOVOLC simulation with respect to the total quantities of species in the CARNALTI simulation:

$$ContributionX = 100 * \frac{X_{CARNALTI} - X_{NOVOLC}}{X_{CARNALTI}} \qquad (4)$$

with $X_{CARNALTI}$ and $X_{NOVOLC}$ the annual mean concentration of the parameter $X$ in CARNALTI and NOVOLC simulations, respectively.

Hereafter, the parameters from NOVOLC simulation will be refered as "non-volcanic" parameters. On the contrary, "volcanic" parameters correspond to the parameters of CARNALTI simulation substracted by NOVOLC simulation. CARNALTI simulation represents the total concentration of the parameters.

**6.1   Global budgets**

The global sulfur budget simulated in CARNALTI is shown in Table 8. Annually and globally averaged SO$_2$ emissions, SO$_2$ and sulfate aerosols burdens as well as sulfur wet and dry depositions are used to calculate the sulfur budget.

The global SO$_2$ burden is 0.17 Tg, smaller than but close to other studies which found values ranging from 0.2 Tg to 0.52 Tg [Pham et al. (1995), Chin and Jacob (1996), Feichter et al. (1996), Graf et al. (1997), Stevenson et al. (2003)]. Stevenson
et al. (2003) explained that the SO$_2$ burden in a simulation can change depending on the distribution of oxidants and the deposition scheme used in the models. Moreover, those studies were done on earlier periods of time when anthropogenic SO$_2$ emissions were superior compared to 2013 in our simulation. In our simulation, 21.97 Tg S are directly removed by dry and wet deposition from sulfur dioxide, representing a percentage of almost 27 %. Thus, the transformation rate to sulfate is about 73 %; a bit higher but consistent with the studies reported above (from 50 to 64 %).
The global vertical sulfate column is 0.82 Tg S, comparable with other studies; 0.53 Tg S in Chin and Jacob (1996), 0.80 Tg S in Pham et al. (1995), 0.78 Tg S in Graf et al. (1997) or 0.81 Tg S in Stevenson et al. (2003).





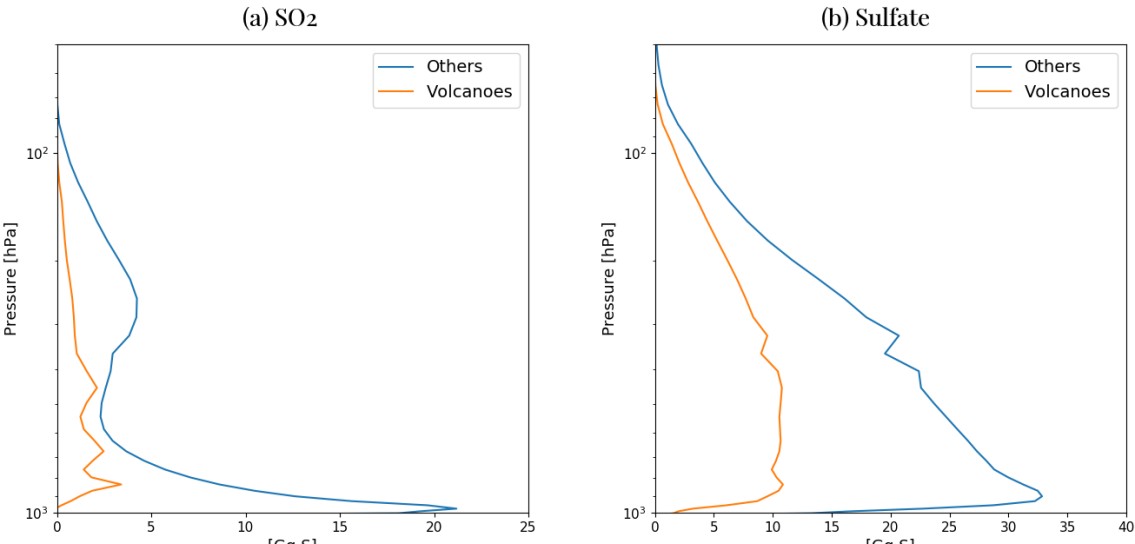

**Figure 8.** *2013 annual global mean vertical profile for (a) SO₂ and (b) sulfate aerosols from volcanic and other sources.*

These results confirm the non-linear contribution of the different SO$_2$ sources to the sulfate burden. Indeed, volcanic sources represent almost 15 % of the total SO$_2$ emitted into the atmosphere, but they contribute 27 % to the sulfate burden. The transformation of SO$_2$ into sulfate from the other sources is not as efficient. We can note a higher efficiency for the volcanic
sources, around 1.89, compared to the other sources, 0.85.

The sulfur deposition is mainly wet deposition. Precisely, the partition of each deposition fluxes are 52 % wet deposits, 35 % dry deposits and 12 % from sedimentation. But, sulfur deposition due to volcanic emissions is weaker than for the other sources: 8 % for wet deposition, 23 % for sulfate aerosol sedimentation and only 2 % for dry deposition. Due to the higher altitude of injection, the atmospheric residence time for sulfur species is longer and the deposition rate lower, especially for the
dry deposition.

### 6.2   Vertical distribution

Figure 8 shows the global and annually averaged vertical profiles for sulfur dioxide and sulfate concentrations for 2013. Anthropogenic and volcanic sources are separated to highlight the main differences between them.

Non-volcanic SO$_2$ dominates the entire vertical column, with a maximum at the surface linked to anthropogenic emissions,
only emitted in MOCAGE first levels. On the contrary, the vertical distribution from volcanic SO$_2$ shows vertical variations. There is no contribution below 950 hPa and two maxima; one at 850 hPa, due to passive degassing and small eruptions, and the other around 630 hPa due to high-altitude eruptions.

However, non volcanic sulfate still dominates the entire vertical column, with the highest values around 950 hPa. For volcanic sulfate vertical distribution, the maximum is between 900 and 750 hPa but four times smaller than for other sources. These





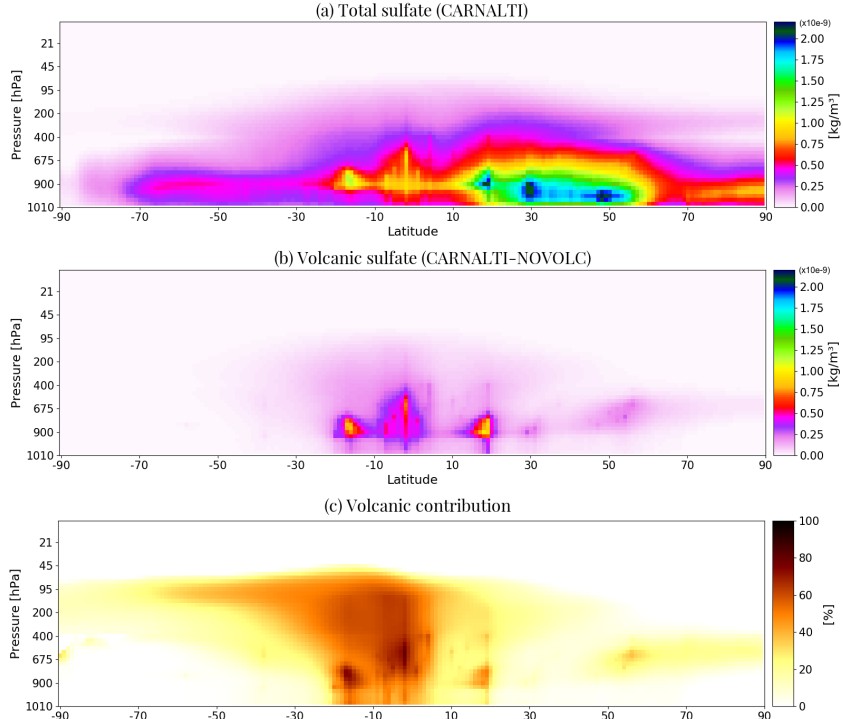

**Figure 9.** *2013 annual zonal mean (a) total sulfate concentration (in kg.m$^{-3}$), (b) volcanic sulfate concentration (in kg.m$^{-3}$) and (c) volcanic sulfate contribution (in %).*

results are different from Graf et al. (1997), which shows that the vertical distribution of volcanic sulfate aerosols is comparable to anthropogenic and biomass burning sulfate and is even dominant between 800 and 200 hPa. This corresponds to the altitude of volcanic emissions in Graf et al. (1997) mainly in the form of eruptions. This can explain the greater efficiency of 2.63 in the tropospheric sulfate burden in Graf et al. (1997). Since volcanic sulfate aerosols are not dominant in the upper troposphere in our simulation, this could explain our lower efficiency for the sulfate burden (1.89 in this study).

Figure 9a represents the annual zonal mean sulfate concentration. Most of the sulfate aerosols reside in the Northen Hemisphere (between 15° N and 55° N) due to anthropogenic influence and the highest values remain below 900 hPa (close to the surface). However, the sulfate concentrations due to volcanic emissions (Fig. 9b) are at higher altitudes. On both sides of the equator, the plumes are between 900 and 650 hPa with a maximum reached at 400 hPa around the equator. Therefore, over the tropical region, the volcanoes contribution to the sulfate aerosol concentrations is the more important, with a maximum of

75-80 % around 650 hPa (see Fig. 9c). We also notice that sulfate aerosols are transported by the general atmospheric circulation, up to the UTLS (Upper Troposhere - Lower Stratosphere) and from the equator to the poles, especially in the Southern Hemisphere where more volcanoes are located.



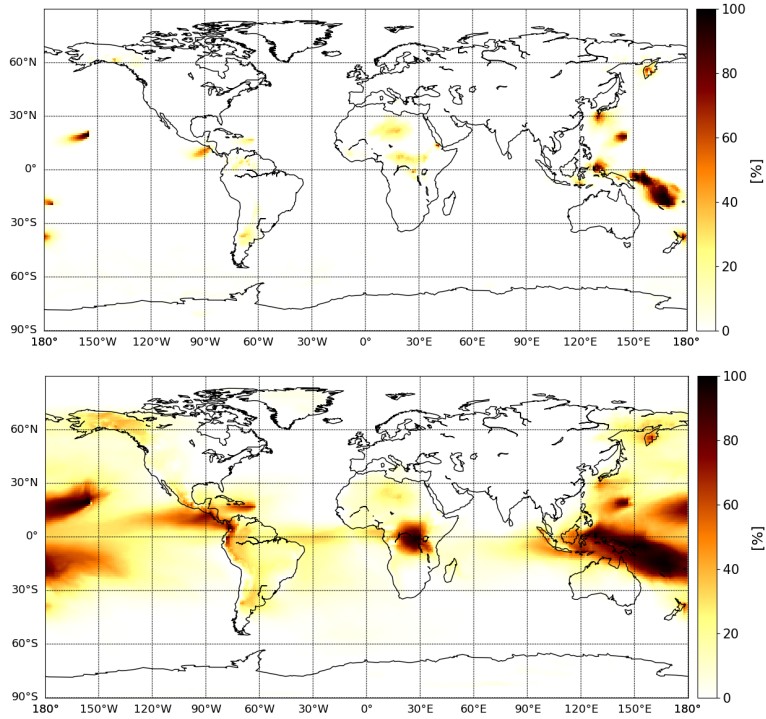

**Figure 10.** *2013 annual mean (top) SO$_2$ and (bottom) sulfate surface contribution due to volcanic emission (in %).*

### 6.3 Regional distributions

The volcanic contribution to the global surface SO$_2$ concentrations is relatively low, around 14 %, but it is much higher close to
the source points (see Fig. 10, top for SO$_2$). This is mainly due to the high altitudes of emissions from volcanoes. In the same
way, we can look at Fig. 10 (bottom for sulfate aerosol) which shows a greater influence of volcanic emissions on the sulfate
aerosol concentration at the surface, almost more important than other sources near the volcanoes. Globally, the mean contribu-
tion is of 19 %, but with a rather low, almost zero, contribution over continental areas in the Northen Hemisphere. Considering
that within the boundary layer, anthropogenic SO$_2$ emissions are dominant, the sulfate aerosols formed in this environment
come largely from anthropogenic sources rather than from the other ones. However, in areas with small anthropogenic sources
(Indonesia, Hawaï and Central Africa), the volcanic contribution is important.

For the total column, volcanic emissions contribute much to the sulfur species burden; 24 % to SO$_2$ and 21 % to sulfate
aerosols. In Fig. 11, we can see that the highest sulfate burden is located over polluted areas (East of North America, Europe,
Middle-East, India and China) as well as near some volcanoes and particulary over oceanic volcanoes. And by looking at
the volcanic contribution, we note that the sulfate aerosols due to volcanic emissions are mainly distributed over the oceanic
environment, near the tropics (also corresponding to volcanoes of lower altitudes). The maximal contribution, 85 %, is found
over Indonesia.




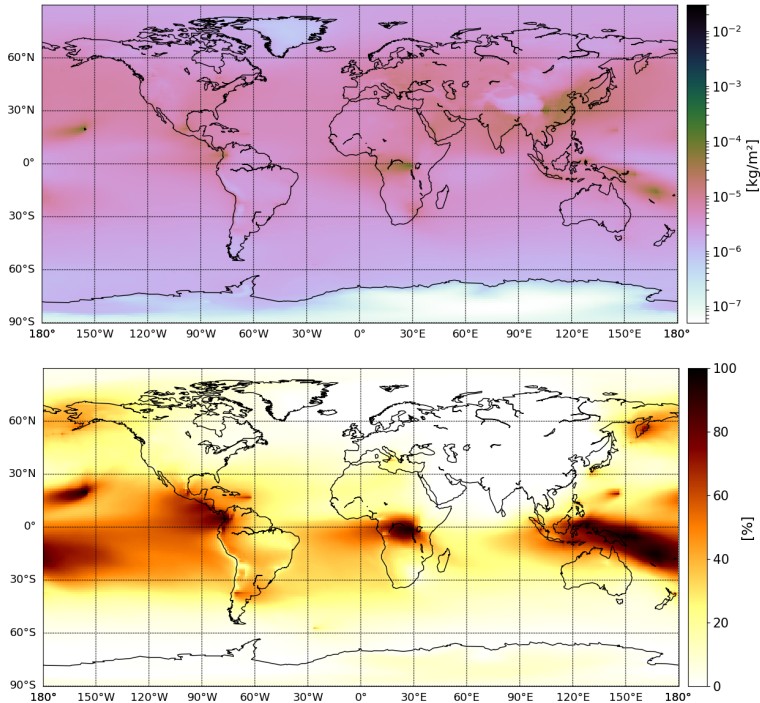

**Figure 11.** *(top) 2013 annual mean sulfate tropospheric column from* CARNALTI *(in kg.m$^{-2}$) and (bottom) its contribution due to volcanic emission (in %).*

The annual global depositions of sulfur species due to volcanic emissions are 20 %, 5 % and 26 % for wet deposition, dry deposition and sedimentation, respectively. Figure 12a represents the total sulfur deposition at the global scale and shows

higher deposition fluxes over anthropogenic polluted areas, where volcanic contribution is low (see Fig. 12b). The only exception area where there is a high deposition flux and a high volcanic contribution is Indonesia. In those areas, we can expect environnemental and health issues due to this pollution. Details on the proportion of each type of deposition (wet, dry and sedimentation) are shown in Fig. S10, where we notice a small flux deposition due to sedimentation, consistent with Table 8, compared to wet and dry depositions.

**7   Sensitivity analysis on passive volcanic source**

Carn et al. (2017) inventory for passive degassing provides not only annual $SO_2$ volcanic emissions ($\overline{E_{V,Y}}$), but also annual emission uncertainties ($u_{V,Y}$) for each volcanic source. Thus, in this section, we aim at using this information to check the variability induced in MOCAGE sulfur budget (from the lowest to the highest estimation of volcanic emissions) and look at how it affects our conclusions from the previous section.





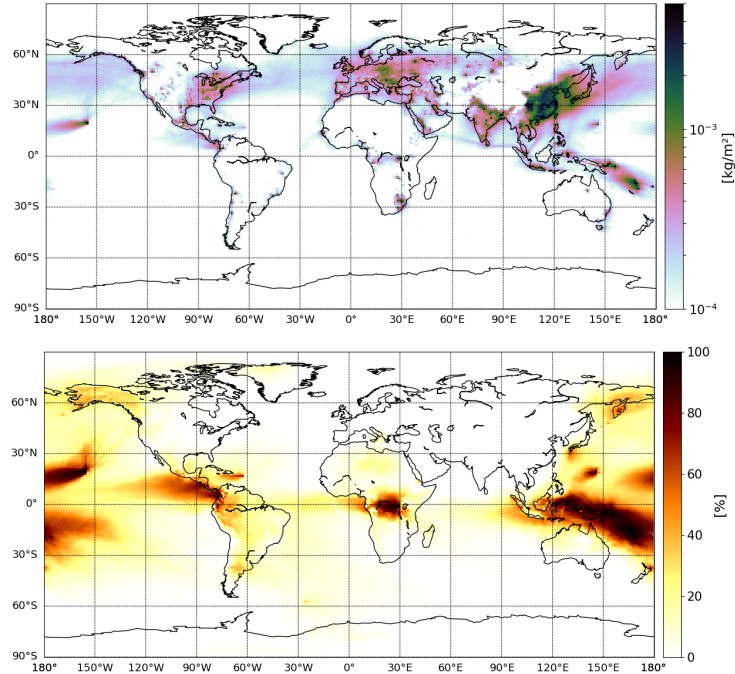

**Figure 12.** *(top) 2013 annual mean sulfur deposition from* CARNALTI *(in kg.m$^{-2}$) and (bottom) its contribution due to volcanic emissions (in %).*

## 7.1 Description of the supplementary simulations

Three additional simulations are conducted to look at the sensitivity in MOCAGE model to the volcanic passive emissions. The first one, named CA_MIN, takes into account for each volcano the lowest estimation of SO$_2$ emission. In other words, for each volcano, we substract the annual emission uncerntainty to the annual mean emission: $E_{V,Y} = \overline{E_{V,Y}} - U_{V,Y}$. On the contrary, the second simulation, named CA_MAX, takes into account the highest estimation of SO$_2$ emission; we add the annual emission uncertainty to the annual mean emission: $E_{V,Y} = \overline{E_{V,Y}} + U_{V,Y}$. Thus, both CA_MIN and CA_MAX experiments do not have daily variations due to passive degassing, but only due to eruptions. For the last one, named CA_RAND, emissions are randomly determined on a daily basis within the annual emission uncertainty interval $[\overline{E_{V,Y}} - U_{V,Y} \, , \, \overline{E_{V,Y}} + U_{V,Y}]$ following a continuous uniform distribution. Thus, daily variations are not only due to eruptions but also to passive degassing, as expected in reality. The reference simulation used is the CARNALTI simulation is named as CA hereafter.

Figure 13 presents the 2013 temporal evolution of SO$_2$ total emission for each simulation. As in Fig. 1, we note the annual variation due to anthropogenic emissions, representing a common basis of 70.58 Tg S.yr$^{-1}$ for each simulation, as well as the daily variation due to eruptions, shown by the large peaks and representing a value of 0.10 Tg S in 2013. Therefore, the differences included in the simulation are only in passive degassing SO$_2$ emissions. In CA simulation, the annual total passive degassing emission is 11.74 Tg S. But in CA_MIN, CA_MAX and CA_RAND experiments, the annual total passive degassing





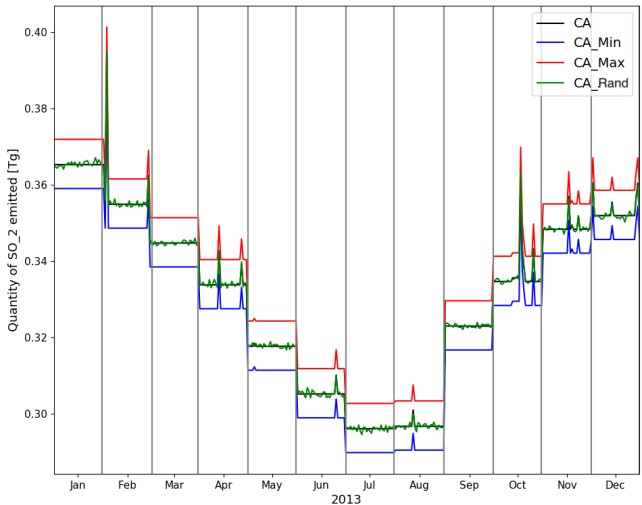

**Figure 13.** *Temporal evolution of 2013 $SO_2$ emissions, corresponding to* CA *(black),* CA_MIN *(blue),* CA_MAX *(red) and* CA_RAND *(green) simulations.*

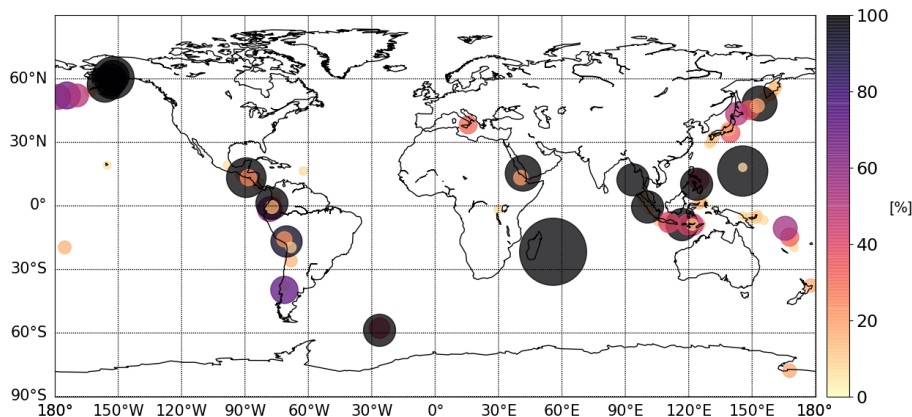

**Figure 14.** *Map of $U_{V,Y}/\overline{E_{V,Y}}$ normalized ratio of $SO_2$ emissions (in %) in Carn et al. (2017) inventory. The size and color of the rounds are proportionnal.*

emission is 10.60, 12.95 and 11.75 Tg S, respectively. Thus, a relative difference of 10.6 % with respect to the annual mean volcanic emissions for CA_MIN simulation is noted; but a difference of 1.42 % when considering all sulfur species emission. Similarly, volcanic emissions in CA_MAX and CA_RAND simulations are 9.3 % and 0.05 % higher than the annual mean volcanic emissions in CA; which represents a difference of 1.47 % and <0.01 % respectively with respect to the total sulfur emissions.

We expect a greater sensitivity to the annual emission uncertainty at volcanoes where the proportion of the annual uncertainty with respect to the annual mean emission is close to 100 %. Figure 14 represents the percentage share of uncertainty on the





**Table 9.** *As in Table 8 but for* CA_MIN*,* CA_MAX *and* CA_RAND *simulations.*

| | | Sulfur Emission | SO$_2$ Burden | Sulfate Burden | Sulfur Deposition | | | Efficiency |
|---|---|---|---|---|---|---|---|---|
| | | | | | Wet | Dry | Sedim | |
| **CA_MIN** | Total | 80.27 | 0.16 | 0.80 | 41.45 | 28.14 | 10.01 | - |
| | *Sources contributions to the total budget (%)* | | | | | | | |
| | Volcanoes | 13.33 | 12.68 | 25.30 | 5.74 | 3.43 | 24.59 | 1.90 |
| | Other | 86.67 | 87.32 | 74.70 | 94.26 | 96.57 | 75.41 | 0.86 |
| **CA_MAX** | Total | 82.62 | 0.17 | 0.84 | 42.75 | 28.38 | 10.48 | - |
| | *Sources contributions to the total budget (%)* | | | | | | | |
| | Volcanoes | 15.80 | 14.87 | 29.02 | 8.61 | 4.27 | 28.01 | 1.84 |
| | Other | 84.20 | 85.13 | 70.98 | 91.39 | 95.73 | 71.99 | 0.84 |
| **CA_RAND** | Total | 81.42 | 0.17 | 0.82 | 42.09 | 28.26 | 10.24 | - |
| | *Sources contributions to the total budget (%)* | | | | | | | |
| | Volcanoes | 14.55 | 13.74 | 27.17 | 7.17 | 3.83 | 26.30 | 1.87 |
| | Other | 85.45 | 86.26 | 72.83 | 92.83 | 96.17 | 73.70 | 0.85 |

annual measurement of volcanic emission per volcano in Carn et al. (2017) inventory. The darker and bigger the circle is, the more important is the uncertainty compared to the mean emission.

## 7.2 Sensitivity study on the global budget in MOCAGE

As in Table 8 for CA, Table 9 presents the annual mean global sulfur budget for CA_MIN, CA_MAX and CA_RAND simulations. Even if the total sulfur species burdens are similar in all simulations with SO$_2$ burden around 0.17 Tg S and sulfate burden between 0.80-0.84 Tg S, we notice that the contribution of volcanic emission varies. In CA experiment, the volcanic contribution to the sulfate aerosol burden is 27.40 %, but it ranges from 25.30 % in CA_MIN to 29.02 % in CA_MAX experiments. This implies a variation in the efficiency of the model MOCAGE to produce sulfate aerosols from volcanic SO$_2$

emissions. The greatest efficiency score is 1.90 for CA_MIN simulation, meaning that smaller amounts of SO$_2$ emitted can form sulfate more efficiently. It illustrates perfectly the non-linear relation between the volcanic SO$_2$ emission and the sulfur budget.

Figures 15 illustrate the spatial difference in volcanic SO$_2$ contribution between CA and the other simulations CA_MIN, CA_MAX and CA_RAND. We note that the differences with CA_MIN or CA_MAX (Fig. 15a and 15b) are similar but of

opposite sign. As expected, difference are both located in the vicinity of volcanic point sources but especially near volcanoes with a high $U_{V,Y}/\overline{E_{V,Y}}$ ratio (see Fig. 14). The contribution of volcanic SO$_2$ is more important (less important resp.) in CA_MAX simulation (CA_MIN simulation resp.) than in CA. Nevertheless, the difference between CA and CA_RAND are weaker. The daily variation in SO$_2$ emissions of volcanoes do not change significantly the annual mean contribution of volcanic SO$_2$ tropospheric column.



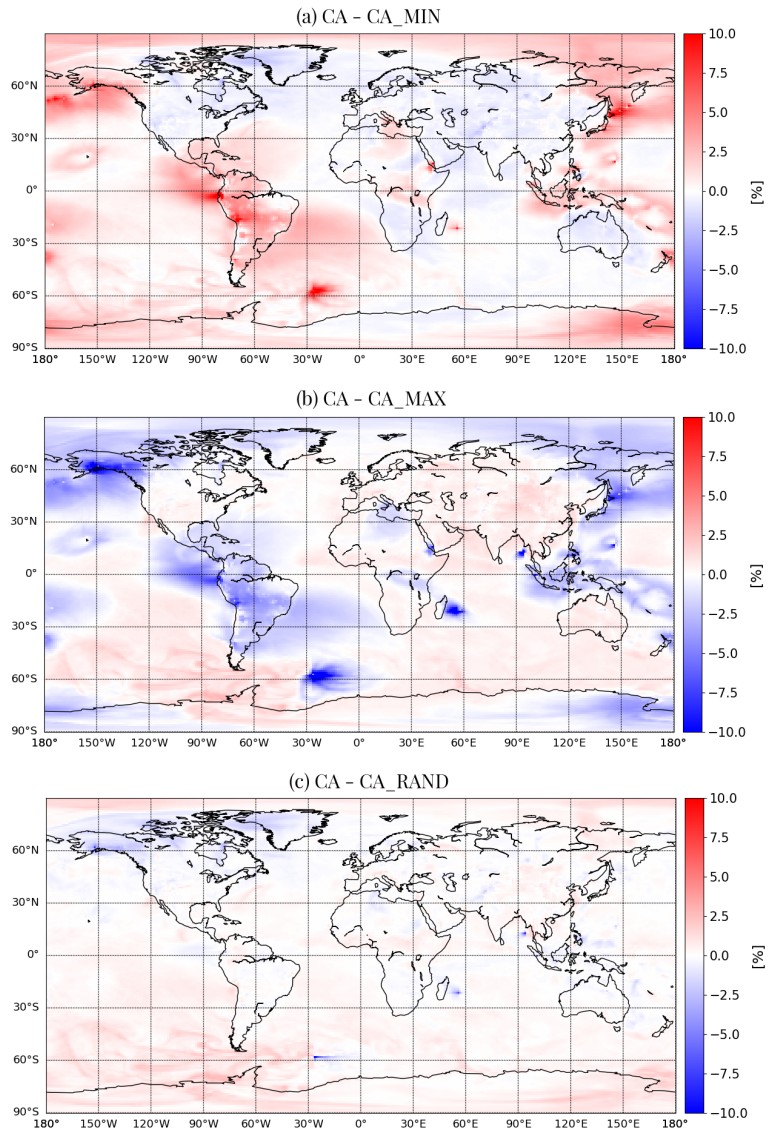

**Figure 15.** *2013 annual mean difference in $SO_2$ tropospheric column volcanic contribution between* CA *and (a)* CA_MIN*, (b)* CA_MAX *and (c)* CA_RAND *simulations, in %.*

Same conclusions are shown in Fig. S11 for sulfate. Yet, the dispersion of sulfate aerosols being stronger, the differences of contribution are greater.

However, the differences between the simulations remain mostly in the deposition fluxes. Regardless of the sensitivity simulation, the quantities of sulfur deposition (by dry deposition or sedimentation) are higher than in the CA simulation. As an example, the sulfur sedimentation is 9.80 Tg in CA simulation but 10.01, 10.48 and 10.24 Tg in CA_MIN, CA_MAX





and CA_RAND simulations, respectively. It represents a contribution of 22.97 % for CA but 24.59, 28.01 and 26.30 % for
CA_MIN, CA_MAX and CA_RAND, respectively. Thus, we can conclude that sulfur deposition reacts inhomogeneously both
to the quantities of volcanic $SO_2$ emitted (with respect to CA_MIN and CA_MAX simulations) and to the spatio-temporal
variability of these emissions (with respect to CA_RAND).

Finally, we can conclude that taking into account the annual emission uncertainties by volcanoes allows us to evaluate
MOCAGE sensibility and variability. In CA_MAX experiment, with the highest estimation of volcanic emissions, we find, as
expected, higher sulfur burden as well as higher sulfur deposition quantities. However, on a reverse approach with CA_MIN
simulation, by assuming the lowest estimate of volcanic $SO_2$ emissions, we do find slighly lower quantity of sulfur deposition,
but with a very different distribution. While wet deposition is lower between CA and CA_MIN, dry deposition and sedimen-
tation are more important in CA_MIN simulation. Even when appling a daily variation, with nearly the same total annual
quantity of volcanic $SO_2$ emitted (simulation CA_RAND), we notice changes in MOCAGE sulfur budget. Thus, it would be
recommended to use an inventory providing daily measurements to correctly simulate the global sulfur budget with MOCAGE.

## 8  Conclusions

In this paper, the aim was to study the contribution of volcanic sulfur emissions on the tropospheric composition and on sulfur
species surface concentration and deposition, at the global scale. Currently, the volcanic emissions inventory implemented in
MOCAGE is from Andres and Kasgnoc (1998), but it has become obsolete. Therefore, a new volcanic $SO_2$ emission inventory,
based on Carn et al. (2016, 2017), is implemented in MOCAGE. Thanks to satellite technologies, used to compile this inventory,
it references more volcanoes and gather both eruptive emissions and passive degassing at a fine time resolution compared to
previous inventories. Eruptions are provided as daily total amounts and passive degassing as annual averages with annual
uncertainties. The inventory also provides information on the altitude plumes. Thus, a parameterization to inject volcanic
emission in altitude with an umbrella profile was implemented in the model.

The choice was made to consider the year 2013, when quantities of volcanic $SO_2$ from eruptions are the lowest in the new
inventory and negligible in the yearly average. Thereby, the study is focused on passive degassing emissions. Four different
simulations are carried out; without (NOVOLC) or with (REF, CARN, CARNALTI) volcanic $SO_2$ emissions and without (REF,
CARN) or with (CARNALTI) volcanic emissions injected in altitude.

The comparison of MOCAGE simulations against GOME-2 $SO_2$ total column and MODIS AOD shows that the statistical
scores of the model were improved in CARNALTI simulation compared to REF, especially at the local scale near the volcanoes.
Hence, well constraining volcanic emission sources in Chemistry-Transport Models (CTM) is necessary, in order to better
represent the tropospheric composition.

Thanks to the four different simulations, we showed that considering more volcanoes (both passive degassing and eruptive
types) and using a parameterization to inject volcanic emissions in altitude allows MOCAGE to increase sulfur species con-
centrations in CARNALTI compared to REF. At the surface, sulfur species concentration and deposition were also increased,




especially in the vicinity of the volcanoes, affecting air quality in these areas possibly causing environmental and health problems.

With this new volcanic emissions inventory, we were able to analyse the model sulfur budget in the troposphere. It was
shown that even if volcanic emissions represents only 15 % of the total sulfur emissions, the contribution of volcanoes to sulfur species is non-linear. Indeed, volcanic sulfate burden is around 27 %, which points out that the volcanoes efficiency to the sulfur budget is greater than from other sources. Similarly, sulfur deposition due to volcanic emissions contributes inequally to the total sulfur deposition, depending on the nature of deposition; e.g. volcanic sulfate aerosols sedimentation represents the smallest proportion of the total volcanic sulfur deposition (about 12 %), but contributes significantly to the total sulfur
sedimentation from all types of $SO_2$ sources (about 23 %).

Moreover, the model sensitivity to passive degassing shows that the variation of volcanic sources is important. By increasing, decreasing or including temporal variation in volcanic emission fluxes, the global sulfur budget changes non-linearly. As an exemple, even if sulfur deposition is slightly lower in CA_MIN than in CA simulation, the dry deposits and sedimentation are higher. Despite a reduction in the amount of volcanic $SO_2$ emitted, the distribution in sulfur deposition varies, causing the
decrease of wet deposition but the increase of dry deposition and sedimentation. These results show that it would be interesting to have more accurate inventories of volcanic emissions with a daily variation. With the constant improvement of space-borned instruments, it is realistic to hope for the availability of such inventories in the coming years.

In this study, we focused on one particular year, and by choosing the 2013 year, we study mainly the impact of passive degassing emissions. However, additionnal studies considering a year when volcanic eruptions are more important would be
complementary; e.g. in 2014, 5.35 Tg of eruptive emissions are referenced, almost thirty times more than in 2013. It would be interesting to compare and analyse the specific impact of eruptive emissions on the tropospheric sulfur budget. However, the comparison of the tropospheric sulfur budget between different years can not only be affected by the differences in volcanic sulfur emissions. Indeed, sulfur dioxide is a soluble species and the meteorological parameters can also impact the tropospheric sulfur budget; e.g. differences in precipitation can lead to changes in the wet deposition fluxes for different years. Thus,
meteorological parameters should be taken into account and inter-annual differences should be studied with cross simulations.

Finally, it could also be interresting to not only compare two years of the Carn et al. (2016, 2017) inventory, but to fully study the inter-annual variability of volcanic sulfur emissions over a longer period. Since the data are fully available over a decade (2005-2015), this type of study would be entirely possible and valuable to scrutinize the tendency of volcanic sulfur emissions contribution.

*Data availability.* The new volcanic $SO_2$ inventory implemented is available for eruptive emissions on GES DISC archive (https://doi.org/10.5067/MEASURES/SO2/DATA404) and for passive degassing emissions on Carn et al. (2017) supplementary information (https://doi.org/10.1038/srep44095). Concerning data used for the validation, GOME-2 $SO_2$ total column data can be find at SAF Archive or EUMET-SAT Data Centre (https://doi.org/10.15770/EUM_SAF_O3M_0013).The current volcanic $SO_2$ inventory is available upon request from the corresponding author.



*Supplement.* The supplement related to this article is available online at: . . .

*Author contributions.* CL, JG and VM designed the study and the model experiments. Simulations were carried by CL with help from JG and MC. The paper was written by CL and reviewed, commented and edited by VM and JG. All authors approved the article.

*Competing interests.* The authors declare that they have no conflict of interest.

*Acknowledgements.* We would like to acknowledge the MODIS mission team and scientists for the production of the data used in this study.
The authors also thank Université Paul Sabatier Toulouse III for funding Claire Lamotte's PhD and Météo-France for hosting it at the Centre National de Recherches Météorologiques.





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
