# Peer review of "Modeling study of the impact of SO2 volcanic passive emissions on the tropospheric sulfur budget"

_Atmospheric Chemistry and Physics, 2020_

## Referee Comment (RC1) · Anonymous Referee #1 · 16 Oct 2020

This modelling study of the impact of non-eruptive volcanic SO2 emissions could have been interesting, given the importance of such emissions for the global sulfate aerosol budget. The authors implement a recently developed volcanic emission inventory (Carn et al.) which represents a significant improvement in comparison with the widely-used GEIA inventory. Those inventories are tested using the global model MOCAGE and evaluated against spaceborne SO2 columns.

Unfortunately, they use GOME-2 SO2 columns from ACSAF, maybe the worst possible choice of SO2 satellite data. OMI SO2 data would have been much more appropriate. The SAF dataset is not even the best GOME-2 dataset. In fact, examination of Figure

S1 shows two things: 1) the filtering applied to the columns has an disproportionate impact on the columns, and 2) the filtered GOME-2 columns (Figure S1b) have a completely unrealistic distribution. Hot spots are found in every very dry areas on Earth including South Africa, Mongolia, Tibet, Central Australia and Western U.S. This is a strong and obvious artefact. The North China Plain shows a weak enhancement, but much less pronounced than Tibet. This is not credible at all. As far as I know, this dataset has not been validated nor has it been used for any scientific investigation. The correlation coefficient between the model and the data is negative (-0.17) on the global scale, and it is insensitive to choice of the emission inventory. I 'm afraid that any conclusion drawn from comparisons with the model are probably useless.

The authors should use a better SO2 dataset. I do not accept the argument that "only instruments different from those used to set the inventories can be selected for an indepedent evaluation". On the contrary, it seems imperative to confront the model with OMI SO2 data and check the overall performance of MOCAGE against those data. It would make the paper much more interesting. Confronting the model with GOME-2 could be interesting as well, but a better dataset would have to be used.

Some additional important comments:

The paper insists several times that "the contribution of volcanic emissions is argued as non-linear on the sulfur species burden". There seems to be quite a confusion regarding the nature of non-linearity. Yes, volcanic SO2 is longer-lived than SO2 from other sources, because it is emitted at higher altitudes and is therefore less subject to dry and wet deposition. But this does not make the contribution of volcanic emissions "non-linear". It would be non-linear if the SO2 emissions would significantly alter their own lifetime (as is the case e.g. for NOx, due to the strong influence of NOx on hydroxyl radical concentrations). I don't think this is what the authors mean here. The emphasis on the role of non-linearity should be removed from the paper.

The paper also insists that non-eruptive volcanic emissions were injected at the first

model level in previous studies. This is not correct. The altitude of the mouth of the volcanoes is of course well known since a long time, and was taken into account already in the early global studies of the sulfur cycle, e.g. Spiro et al. (1992), Pham et al. (1995), Chin et al. (2000). The crater lies generally much higher than the lowest level of the model.

Finally, the article is difficult to read. The text is often unnecessarily long, with many repetitions. The English language could be much improved, see below my suggestions for changes. Many sentences are either illogical, confusing or of little informative value. It is not normal that a reviewer would have to correct so much a scientific paper.

Minor comments

l. 32 COSPEC: here, make reference to section 3.1 which explains what it is.

l. 33 TOMS: make clear that TOMS provided only crude measurements of SO2 columns.

l. 117 "first five levels": indicate the approximate altitude range. Why not injecting emissions at the first level only?

l. 142-144 The description of SOA parameterization is very brief, and could be expanded. How well does it perform against organic aerosol observations?

Table 1 states that the Carn et al. inventory relies on TOMS and OMI, whereas the text mentioned 7 different satellite instruments.

l. 253-254 "One simulation takes into account only anthropogenic emissions": strange, no biomass burning or natural S emissions? –> replace by "The first run (NOVOLC) neglects volcanic emissions." Adapt also the rest of the paragraph.

l. 291 "daily mean SO2": satellites do not provide daily means.

l. 295 "thanks for fitting AMF": unclear. As far as I know, the AMF is not fitted.

Table 3 does not bring much, since the global MNMB is given in the text, and the correlation coefficient is negative. Could be moved to the supplement. The global MNMB is not much interesting given the compensation between very high and very low values apparent in Figure 3.

l. 360 "We notice small changes in the vicinity of volcanoes where MNMB score is improved": there are many cases where the MNMB is worsened, including Hawaii and islands (Vanuatu?) in the Southern Pacific.

l. 361 "FGE is better" –> "The FGE is slightly improved"

l. 362 Some comments are needed concerning the negative value of the correlation coefficient.

At this point, I stopped reading, since I think that the GOME-2 dataset (on which this investigation relies) has no scientific value.

Minor (language) comments

l. 1 Why "Thus"? The sentence remains true even in absence of non-linear behavior.

l. 3 at the global surface –> at the global scale (?)

l. 4-5 I would rephrase as "the changes induced by the update of the volcanic emissions inventory are studied using the ..."

l. 7 "degassing" –> "degassing emissions"

l. 8 "uncertainties by volcanoes": what does that mean?

l. 9 "negligible"

l. 12 and elsewhere: remove the dot between Tg and yr

l. 17 "necessity of estimates accurate volcanic sources" –> "need for accurate estimates of volcanic sources"

l. 20 delete "natural" before "volcanic"

l. 25 "Plus" –> "Moreover"

l. 30 "to well constrain" : ?

l. 43 "were not very accurate in quantitative, spatial and temporal detection": weird wording, please rephrase

l. 44 "used on": ??

l. 46 Andres and Kasgnoc (1998) work –> The study of Andres and Kasgnoc (1998)

l. 51 "As well": ??

l. 55 "in its works": ??. "more numerous and qualitative data": ??

l. 59 "for passive source strength": ??

l. 61 "high change" –> "stark improvement"

l. 61 "last decades studies" -> "studies of the last decades"

l. 63 "the radiative forcing induced" -> "the subsequent radiative forcing"

l. 67 "on surface species concentration and deposition" -> "on the surface concentration and deposition of sulfur species"

l. 67 "We want": ??

l. 74 the configuration of simulations with MOCAGE

l. 76 "updating inventory": ??

l. 76 "the comparison for"

l. 77 "Then" –> "Next"

l. 83 "Its use is applied": ??

l. 94 "the duration emission" -> "the duration of the emission"

l. 115 "from biomass burning process" –> "emitted from biomass burning"

l. 124 "completed": ??

l. 158 "It was carried out over a period of about 25 years": I suppose you mean the measurements span 25 years. Please rephrase.

l. 167 "thanks to the similar molecular structure of SO2 and ozone": misleading, rephrase or omit.

l. 167 "Thus": ?? The following sentence is unclear. This could be simplified, as not really necessary.

l. 178 "constancy"–> stability

l. 178 "Thus" could be omitted

l. 178-179 "in order to incorporate natural variations due to temporal and even chemical inhomogeneities": confusing. Could be omitted.

l. 181 "as the one...": replace by "as being among the largest..."

l. 181 "passive": ??

l. 181 "For them..." -> "For those volcanoes, fluxes (...) supersede the averages"

l. 185 "Knowing that" –> "Since"

l. 189 "lowest levels" or "lowest level"?

l. 191 "technological improvements in satellite technology": awkward.

l. 197 The work of Carn et al. (...) updates and completes the study of Andres and Kasgnoc (1998).

l. 203 "given is" –> "given includes"

l. 204 "measured" –> "estimated"

l. 205 "We will..." is somewhat ambiguous. Within this study or later on?

l. 207 "the daily frequency allows to take into account the eruptions in simulations for the period...": weird statement.

l. 211 "could distinguish" –> "made possible to distinguish"

l. 222 "every day of the year" -> "throughout the year"

l. 236 "the update of the..." –> "the updated..."

l. 236 "compiles" -> "includes"

l. 237 "spread over the globe": ??

l. 244 Delete words "lon" and "lat"

l. 246 "volcanic one" –> "volcanic source"

l. 246 "The same global annual sulfur emissions are computed for all other sources": of course since the same inventories are used!

l. 248 "emissions are" -> "emissions amount to". You don't need two significant digits after the decimal point, one is enough.

Table 1 legend "Summary information on" –> "Summary of"

Table 1 "Nb of volcano" –> "Number of volcanoes"

l. 253 "characteristics" –> "main features"

l. 256 "However, one injects the volcanic SO2 emissions" -> "In simulation CARN, volcanic emissions are injected". Adapt also the rest of the sentence

l. 261 "in altitude" –> "in the vertical"

l. 262 "Then": ?? The entire sentence is weird. You could dropt it since you explain

what you do in thefollowing sentences.

l. 263 "The CARNALTI run is expected to provide the best..."

Figure 1 legend: drop "annual" (since monthly values are shown). "anthropogenic" or "other emissions"?

l. 269 lowest eruptive emission flux (Carn et al., 2016)

l. 269 "is negligible". That sentence could be dropped.

l. 270 Why the upper-case AND?

l. 270 This sentence is weird, not really useful.

l. 273 "adds": ??

l. 275 "referenced": ??

l. 276 "counts": ??; "into": ??

l. 277 "are" –> "amount to". Use only one significant digit for the totals.

l. 279 "current": the use of this word for the previously used inventory is weird. Replace maybe (here and elsewhere) by "previous"

l. 280 "inventory against" –> "not accounted for by". Delete "one".

Figure 2 Legend "rounds" –> "circles"

l. 284-285 Weird sentence, provide more direct formulation.

l. 285 "benefit". The sentence is true but too obvious.

l. 289 "indirectly correlated to SO2": ??

l. 299 "Plus" -> "In addition"; "presence of offsets" –> "offsets" (?); "lead" –> "leads"; "criteria" is plural, replace by "criterion" (if meant as singular)

l. 301 "subtracted at" –> "subtracted from"

l. 311 "low confident" –> "low-confidence"

l. 311 "filtered" -> "filtered out"

l. 318 "we can use several statistical metrics": delete, and merge with next sentence "we use the fractional bias,..."

l. 335-337 This paragraph could be omitted. Delete "Therefore" from the next paragraph.

l. 340 "Plus" -> "Furthermore". But I don't understand well the rest of the sentence. Rephrase.

l. 343 You might drop the word "inventory" after the reference. Same remark applies elsewhere in the text.

l. 342 Drop "The" before Zone 1. Same elsewhere.

l. 346 "are" -> "amount to"

l. 354 "counting"??

l. 358 "higher" –> "less negative"

l. 359 "againts" -> "against"

Table 4 "Coorelation" –> "Correlation"; "specifics" -> "specific"

l. 698: the link does not work

REFERENCES

Chin, M., Rood, R. B., Lin, S.-J., Muller, J.-F., and Thompson, A. M.: Atmospheric sulfur cycle simulated in the global model GOCART: Model description and global properties, J. Geophys. Res. 105, 24671-24687, 2000.

Pham, M., Muller, J. F., Brasseur, G. P., Granier, C., and Megie, G.: A three-dimensional study of the tropospheric sulfur cycle, J. Geophys. Res. 100, 26061-20092, 1995.

Spiro, P. A., Jacob, D. J. and Logan, J. A.: Global inventory of sulfur emissions with 1°x1° resolution, J. Geophys. Res. 97, 6023-6036, 1992.

---

## Short Comment (SC1) · 2 Dec 2020

Dear Authors,

I have seen your manuscript "Modeling study of the impact of SO2 volcanic passive emissions on the tropospheric sulfur budget" in ACPD and it captured my interest. Unfortunately, while reading the manuscript, I have found a number of possible major flaws that I would like to mention during this Discussion phase. Please find it in the following.

The manuscript introduces a new and more detailed volcanic emission inventory (by

Simon Carn), input to MOCAGE CTM modelling, and evaluates the improvements brought in the global and regional sulphur budget with respect to older inventories using satellite observations as reference. The topics of this manuscript is of certain interest for multiple communities (atmospheric modellers, atmospheric scientists, climate scientists and volcanologists) and is worth attention. Unfortunately, I have found the following major flaws that, in my opinion, invalidate the results of this work that I think should not be published in the present form. I'll be available to review a possible revised manuscript in case the Editor thinks it useful.

My best regards,

Pasquale Sellitto

Major comments:

1) In the introduction lines of Sect. 3 (L150-155), it is said that SO2 is the main volcanic effluent and is the only volcanic emission considered in this work. This is absolutely not true. The single most important volcanic effluent is not SO2 but water vapour, with water vapour/SO2 emitted mass ratios reaching values as large as a few hundreds. I think that many emissions and near-source volcanic processes can safely be neglected, as a first approximation, like halogen emissions and their impact on sulphate formation, transition metal contribution and other interactions of SO2/sulphate with ash including heterogeneous chemistry; nevertheless, volcanic water vapour emissions cannot absolutely be neglected, as well as their in-plume effects on sulphate formation and SO2 depletion. In my perspective, neglecting water vapour emissions (as said, the dominating gaseous effluent in volcanic degassing) invalidate the results of this work.

2) I agree with Referee #1 on the fact that ACSAF GOME-2 retrievals are not a good choice for the validation of the MOCAGE simulations. I'd also mention that, differently to what said at L287, GOME-2 data are not completely independent on the OMI and TOMS input data to your inventories: the three instruments operate in the UV spectral range and use similar spectral ranges and SO2 absorption structures for the retrieval.

Why not using infrared instruments as IASI, instead?

3) Also, the choice of MODIS AOD is quite debatable strategy. MODIS AOD is linked to all aerosols, not only sulphates. how do you separate sulphate aerosols from the other aerosol typologies/composition/sources? For example, in "Region 3 (Mediterranean)" dust is, on average, overwhelmingly dominant with respect to sulphate aerosols: how can you check the improvement of volcanic SO2 sources in such an environment, due to the expected small sulphate signal?

4) In addition, how to interpret the results of the comparisons in Tables 3 and 4? Am I wrong to say that observations and simulations compare very weakly? This is also the case if looking at the (necessary in the main text) Figure S1. The simulations and the observations seem to not describing the same SO2 fields. Results for aerosols compare better but, in my opinion, only because the aerosol fields are dominated by other aerosols (and MODIS is more sensitive to higher altitudes aerosols than boundary layer aerosols, so again probably large dust plumes lofted by convection).

5) In general, the manuscript is poorly written and needs a thorough linguistic review. The description of the scientific context is quite approximative and a lot of key references are lacking – please see specific comments in the following.

I also add here specific and technical comments so to help improving the manuscript.

1) L1: "The contribution…": what contribution?

2) L9: "negliable": "negligible"?

3) L20: Here and all the following discussions (including comparisons with your results): recent assessments of sulphur budget should be discussed here, like (for volcanically quiescent conditions, so of large interest for your study):

- Sheng, J.-X., Weisenstein, D. K., Luo, B.-P., Rozanov, E., Stenke, A., Anet, J., Bingemer, H., and Peter, T. (2015), Global atmospheric sulfur budget under volcanically quiescent conditions: Aerosol-chemistry-climate model predictions and validation, J.

Geophys. Res. Atmos., 120, 256– 276, doi:10.1002/2014JD021985.

4) Also, for climate impacts, this should be cited and possibly discussed:

-Kremser, S., et al. (2016), Stratospheric aerosol-Observations, processes, and impact on climate, Rev. Geophys., 54, 278– 335, doi:10.1002/2015RG000511.

5) L22: "variation of climate": You mean "climate forcing"? (in this case, please specify that you're not talking about SO2 but sulphate aerosols)

6) L23-24: "SO2 emissions had become a major concern in environmental policies, leading to strong reductions in anthropogenic emissions in recent decades." : Not everywhere. Please differentiate geographically between decreasing, stationary and increasing emissions regions and add a reference.

7) L24-25: "Thus, the relative proportion of volcanoes in the total sulfur emission sources tends to increase.": Due to different regional trends of anthropogenic emissions, this statement sounds just arbitrary (unless you have specific references that I don't know)

8) L26-27: "is greater in altitude": you might mean: "increases with altitude"

9) L27-28: "Thus, we now…emissions": not clear, please rephrase

10) L27: "longer": "for longer time periods"?

11) L29: "these variations": which variations?

12) L44-45: Please change the phrasing here: there are very few "easy-to-access" volcanoes (Masaya can be mentioned, maybe), while normally the internal processes themselves build "uneasy-to-access" morphological structures for volcanoes.

13) L51: "information on injection altitude is available": The information on the altitude is still very limited. These are observing systems that have a few units of Degrees of Freedom in vertical profile observations of SO2, mostly between 1 and 2.0-2.5, so not

allowing for detailed altitude information. Please mention this in the text and smooth this statement.

14) L56: what do you mean with "more numerous and qualitative data"?

15) L63-65: "In contrast, few studies focus on the impact on tropospheric composition including air quality, with the exception of case studies of volcanic eruptions...": This is not true. Please look at the following papers of my research group, that aimed at the impact of volcanic activity, including passive degassing of selected volcanoes, on the tropospheric composition and air quality, and the many references therein:

- Sellitto, P., Zanetel, C., di Sarra, A. et al., The impact of Mount Etna sulfur emissions on the atmospheric composition and aerosol properties in the central Mediterranean: A statistical analysis over the period 2000–2013 based on observations and Lagrangian modelling, Atmospheric Environment, Volume 148, 2017, Pages 77-88, https://doi.org/10.1016/j.atmosenv.2016.10.032.

- Sellitto, P., Salerno, G., La Spina, A. et al. Small-scale volcanic aerosols variability, processes and direct radiative impact at Mount Etna during the EPL-RADIO campaigns. Sci Rep 10, 15224 (2020). https://doi.org/10.1038/s41598-020-71635-1

Please correct the wrong statement and cite the previous work mentioned above.

16) Section 2.5: What about the vertical transport, which can pose problems for the modelling of confined plumes, like volcanic plumes, and is discussed in the following paper?

- Lachatre, M., Mailler, S., Menut, L., et al., New strategies for vertical transport in chemistry transport models: application to the case of the Mount Etna eruption on 18 March 2012 with CHIMERE v2017r4, Geosci. Model Dev., 13, 5707–5723, https://doi.org/10.5194/gmd-13-5707-2020, 2020.

17) Section 3: see Major Comment 1

18) L165: "calm": What do you mean with "calm"? "Non-eruptive"?

19) L167: the reference to molecular structure sounds strange here. You might want to say that "SO2 and ozone have absorption bands at overlapping spectral regions" (which is linked to molecular structure) or something like this.

20) L181: "...as one of the largest passive emitters.": Clumsy phrasing. Please rephrase.

21) L181-182: Please add details on the sources of these flux information.

22) L183-184: This is very unclear. Please clarify.

23) L188-189: You mean that volcanic SO2 is emitted at the surface (including orography)? Is orography "smoothed" by the average in-grid topology? This aspect is very important e.g. for Etna. Even in case of passive degassing, its emissions are released at, at least, 3000 m altitude and episodic eruptions can reach, for Etna and Kilauea, quite higher altitudes.

24) Section 3.2: there are many repetitions. In general, all the paper should be condensed and repetitions should be suppressed.

25) L224-226: "We implemented...emissions": Why this parameterisation is not described in details here? How it compares to established parameterisations like the one of: ?

- Mastin, L. G. (2014), Testing the accuracy of a 1-D volcanic plume model in estimating mass eruption rate, J. Geophys. Res. Atmos., 119, 2474-2495, doi:10.1002/2013JD020604.

26) L239: "Finally, the availability of emission heights in this inventory gives a better description of the emission.": At this point I think it is necessary to discuss the limitations in the vertical characterisation of volcanic emissions in the new inventory and the satellite observations used to build it, so to not oversell your new simulations

27) Figure 1 and most figures: Please use larger text and labels.

28) L269: "lowest eruptive…negligible in 2013": How much this is "low"? Is it really negligible? How do you qualify this as "negligible"?

29) L272: reference to summer and winter: Please correct to "northern hemisphere summer/winter" and adapt the discussion.

30) L284-285: ""Due" and "since" in the same sentence is quite clumsy. Please rephrase."

31) Section 4: see Major Comments 2-3.

32) Section 4.2.2 title: "MODIS Aerosol Optical Depth"

33) L349: Please check altitude of Mount Etna, this is not the right altitude.

34) Section 5: see Major Comment 4

35) Section 5: It looks like some of the Figures in the Supplements are needed here in the main text, e.g. S1

36) L430: "(industries…": and dust, of course

37) Figure 8: This figure would be largely more useful with an altitude vertical axis (instead of pressure).

38) L517: "This corresponds…eruption": This is quite straightforward interpretation of these results, but it is important to stress the fact that 2013 is not a "normal" year as even a small number of explosive volcanic eruptions can change the vertical distributions of Figure 8 at the global scale. This has to be discussed and the limits of your simulation (a" predominantly passive degassing" year) must be clearly stated.

---

## Referee Comment (RC2) · Anonymous Referee #2 · 10 Dec 2020

The paper by Lamotte et al. studies the effect of SO2 volcanic degassing emissions on tropospheric sulfur budget, using the MOCAGE global CTM implemented with new volcanic emission inventory of Carn et al. (2016, 2017). By model sensitivity tests, tropospheric SO2, sulfate and AODs simulated with new and old emission inventories for the year 2013 are compared and validated against SO2 GOME-2 and AOD MODIS satellite data sets. The results show that the new inventory (CARNALTI) is the best with reference to satellite observations. Such kind of study is interesting and should provide important information for understanding the sulfur global budget. On the other hand, in the opinion of this referee, the manuscript still needs improving a lot before acceptance for publication.

[Figure]

I agree with Referee #1 and Dr. Pasquale Sellitto that other observational data should be used for the validation of the model simulations. In addition, I am very concerned about the effects of the volcanic emission heights on the simulated results. Tables 1 and 2 present different simulation scenarios, but I missed the detailed information on emission heights of the new inventory of Carn et al. (2016, 2017). As I understand, the emission heights are provided in the work of Carn et al. (2016, 2017), but for the sensitivity test, all the volcanic emissions (eruptive and passive) were arbitrarily forced to the model surface for the scenario CARN. While the SO2 tropospheric columns are compared, can the vertical distributions of SO2, sulfate or aerosols be better simulated using CARNALTI than with CARN?

Other issues:

2.3 Emissions (L109-L114): The emission inventories (MACCity and GFAS) are for the years before 2010, earlier than the simulated year 2013. Can this affect the comparing results?

2.4.1 Gaseous species (L123-125): Two schemes, RACM and REPROBUS, are used for tropospheric chemistry and stratospheric chemistry, respectively. How the model grid cells are distinguished between the troposphere and the stratosphere so that only one of them is applied? How are stratospheric tracers (e.g., CFCs and OCS) and tropospheric tracers (e.g., NMVOCs) treated in the model grid cells? Can TUV calculate the photodissociation rates of stratospheric chemical tracers?

---

## Author Comment (AC1) · 25 May 2021

**Answers to the interactive comments on "Modeling study of the impact of SO$_2$ volcanic passive emissions on the tropospheric sulfur budget" by Claire Lamotte et al.**

Comments on Anonymous Referee #1

We would like to thank the Anonymous Referee #1 for their comments that helped improving the paper.
Our response is organised as follows. After each referee's comment (in italic black font) can be found the authors' response (in normal black font), and where needed, the changes made in the manuscript (in blue). In the revised version of the paper, only the significant changes have been coloured in blue to help identifying any new important improvement.
Also to improve the clarity of the paper and following the referees' comments, we have slightly changed the organisation of the paper by splitting section 5 into two. The new Section 5 is only devoted to the evaluation (ex-Sect. 5.1). Section 6 is on the impact of the inventory update on the species concentrations (ex-Sect. 5.2). Also, the purpose of the CARN simulation was not very clear for the referees. This simulation is only used to understand the effect of improving the altitude of injection. This is why CARN results are only used now for the analysis of the species concentrations in the new section 6 (ex-Sect. 5.2). The manuscript has been revised accordingly.
Please note that the revised manuscript has been read and corrected by an English native speaker and that we have added co-authors to the paper that contributed to the responses to the referees and to the revised version.

Major comments:

*This modelling study of the impact of non-eruptive volcanic SO$_2$ emissions could have been interesting, given the importance of such emissions for the global sulfate aerosol budget. The authors implement a recently developed volcanic emission inventory (Carn et al.) which represents a significant improvement in comparison with the widely-used GEIA inventory. Those inventories are tested using the global model MOCAGE and evaluated against spaceborne SO$_2$ columns. The correlation coefficient between the model and the data is negative (-0.17) on the global scale, and it is insensitive to choice of the emission inventory. I'm afraid that any conclusion drawn from comparisons with the model are probably useless.*

*1) Unfortunately, they use GOME-2 SO$_2$ columns from ACSAF, maybe the worst possible choice of SO$_2$ satellite data. OMI SO$_2$ data would have been much more appropriate. The SAF dataset is not even the best GOME-2 dataset. In fact, examination of Figure S1 shows two things: 1) the filtering applied to the columns has an disproportionate impact on the columns, and 2) the filtered GOME-2 columns (Figure S1b) have a completely unrealistic distribution. Hot spots are found in every very dry areas on Earth including South Africa, Mongolia, Tibet, Central Australia and Western U.S. This is a strong and obvious artefact. The North China Plain shows a weak enhancement, but much less pronounced than Tibet. This is not credible at all. As far as I know, this dataset has not been validated nor has it been used for any specific investigation.*

We agree with the reviewer that the Metop-A GOME-2 SO$_2$ columns presented show unrealistic features in some regions. Not being experts on satellite observations, we had chosen for the model evaluation to use GOME-2 MetopA SO$_2$ columns from DLR provided by ACSAF (ex- O3F-SAF) because those data provide an independent measurement of SO$_2$ with respect to OMI (used in the volcanic emission inventory). Indeed, these data present artefacts and noise. Although we had applied filtering, this was not enough to remove all the unrealistic features. This is probably the reason why these data were mainly used in the literature not at the global scale but on case studies at the regional and local scales [Rix et al (2009,2012), Koukouli et al (2015)], and to detect very large emission sources [Fioletov et al (2013)]. Note that we also investigated the use of GOME-2 MetopB SO$_2$ columns from DLR by ACSAF (ex- O3F-SAF) but the results showed similar unrealistic features in some regions as in GOME-2 MetopB SO$_2$ columns.
All this has lead us to change our evaluation strategy to base it on OMI products as suggested by Referee #1.

*2) The author should use a better SO$_2$ dataset. I do not accept the argument that "only instruments different from those used to set the inventories can be selected for an independent evaluation". On the contrary, it seems imperative to confront the model with OMI SO$_2$ data and check the overall performance of MOCAGE against those data. It would make the paper much interesting. Confronting the model with GOME-2 could be interesting as well, but a better dataset would have to be used.*

As suggested, we choose in the revised version to use OMI SO$_2$ columns data for the model evaluation. We

also changed the approach chosen for the statistical evaluation based on the analysis of the literature. Section 4.2 "Observations used for the evaluation of the simulations" and 5.1 "Evaluation of the simulations" were rewritten to explain our new model evaluation strategy and associated results. Here are the main modifications written in the revised paper:

As for all satellite derived products, the relative uncertainties on $SO_2$ columns are large where the $SO_2$ signal is low, in particular for background $SO_2$ conditions. This is why in the literature, the $SO_2$ satellite comparisons or the model evaluations focus on specific areas close to $SO_2$ sources [*e.g.* He et al. (2012), Fioletov et al. (2013), Wang and Wang (2020)]. Similarly to these studies, our new strategy is to perform the model evaluation only in the vicinity of the volcanic sources. For each volcano, we select 9 model grid points (representing a square of 3°longitude x 3°latitude) with the middle point being where the volcano is located. Altogether it corresponds to 633 points. The mask is applied on each daily OMI $SO_2$ total column measurements and then we perform an annual average for each of the 633 data points. Similarly to the above mentioned studies, the results are shown as scatter plots and the statistical metrics used are the correlation coefficient and the RMSE.

There are various products available in the OMI dataset since OMI instrument has a variable sensitivity depending on altitude and the retrieval of $SO_2$ requires the use of an *a priori* profile. We choose the OMI total column density constrained by the *a priori* profiles from GEOS-5 global model. To test if the evaluation is sensitive to this choice, we use another approach which consists in an interpolation from the altitude where the volcanic emissions are injected in MOCAGE to OMI products for the boundary layer, the low troposphere and the middle troposphere. More precisely, the OMI products PBL, TRL and TRM are used. They correspond to $SO_2$ vertical column density with an *a priori* profile assuming fixed mixing ratio within the planetary boundary layer (around 1 km), lower troposphere (around 3 km) and middle troposphere (around 8 km), respectively. Depending on the altitude of the emissions in MOCAGE, either PBL and TRL, or TRL and TRM, are used for the interpolation.

The interpolation that we made is simple. We could have used the product "Scattering Weight" similar to an averaging kernel (provides information on the vertical distribution of $SO_2$) to made a better validation of our model total column with the observation. However, this method is more complicated to do since it is necessary to pre-process the observation data, adapt them into the validation process in MOCAGE and re-run the simulations.

The comparison between the model and OMI $SO_2$ columns clearly shows an improvement of the model performances in the CARNALTI simulation (see Fig. 1).

3) *The paper insists several times that "the contribution of volcanic emissions is argued as non-linear on the sulfur species burden". There seems to be quite a confusion regarding the nature of non-linearity. Yes, volcanic $SO_2$ is longer-lived than $SO_2$ from other sources, because it is emitted at higher altitudes and is therefore less subject to dry and wet deposition. But this does not make the contribution of volcanic emissions "non-linear". It would be non-linear if the $SO_2$ emissions would significantly alter their own lifetime (as is the case e.g. fro NOx, due to the strong influence of NOx on hydroxyl radical concentrations). I don't think this is what the authors mean here. The emphasis on the role of non-linearity should be removed from the paper.*

The meaning of this statement which was not clear is that, the contribution of volcanic emissions is argued as non-linear with respect to the volcanic sulfur emissions. In other word, by emitting 15 % of volcanic sulfur, we do not find 15 % of sulfur burden in the atmosphere due to volcanic emissions. The use of the word "non-linearity" in the paper was referring to the term used in Graf et al (1997): "The most striking feature is that the contributions of the different sources to the $SO_2$ as well as to the sulfate burden are not linear with respect to their source strengths." In the paper, we were not precise enough because we did not refer clearly to the non-linearity as the non-linearity with respect to the emissions. This has been made clear in the revised manuscript.

4) *The paper also insists that non-eruptive volcanic emissions were injected at the first model level in previous studies. This is not correct. The altitude of the mouth of the volcanoes is of course well known, since a long time, and was taken into account already in the early global studies of the sulfur cycle, e.g. Spiro et al (1992), Pham et al (1995), Chin et al (2000). The crater lies generally much higher than the lowest level of the model.*

It seems that this point was not clear in the paper. We did not intend to emphasise that in previous studies non-eruptive emissions were injected on the first model level. We were only referring to the previous versions of the MOCAGE model, in which non-eruptive emissions were injected on the first model levels. The information on the actual altitude of the volcano vent was not taken into account previously even if often much higher than the model orography (which is by definition a weighted average over the 1° x 1°grid box). We knew this was a weakness of the model. We made this clearer in the revised version of the paper.

Minor comments:

(a) Column_Amount_SO$_2$

[Figure]

(b) Interpolation at the model level of volcanic emission injection

Figure 1: *Scatter plots of annual mean OMI SO$_2$ versus MOCAGE simulations (left: REF, right: CARNALTI) (a) considering total columns and (b) interpolating at the model level where volcanic emissions are injected. Also shown on the scatter plot are 1:1 line (solid grey), linear regression line (black dash), linear regression formula, correlation coefficient (R), root mean squared error (RMSE), number of collocated pairs (N), OMI mean and standard deviation in DU (x), MOCAGE mean and standard deviation in DU (y), and density of collocated pairs (colorbar).*

L-32) *COSPEC: here, make reference to section 3.1 which explains what it is.*

The reference to section 3.1 for COSPEC description has been added.

L-33) *TOMS: make clear that TOMS provided only crude measurements of SO$_2$ columns.*

TOMS was the first satellite instrument to measure SO$_2$ total column from space, and at this time, the instrument specifications and the retrieval algorithms were not providing SO$_2$ estimates as accurate as nowadays. We included in the revised version a piece of text on the TOMS early-days measurements of SO$_2$ in the paper.

L-117) *"first five levels: indicate the approximate altitude range. Why not injecting emissions at the first level only?*

For numerical reasons, in particular linked to the use of a semi-Lagrangian scheme for the tracer advection, it is not recommended to inject strong and localised emissions on a single level in the model. Therefore, the injection is prescribed on the first five levels (from the model surface up to approximately 500 m), but with an exponential decrease. This leads to around 50 % injected on the first level, 25 % on the second level and the remaining mass above. The sentence "The injection profile of anthropogenic and biogenic emissions follows an exponential decrease from the surface level of the model: $\delta_L = 0.5\delta_{L+1}$, with $\delta_L$ the injection fraction of the mass emitted at the level L of the model; meaning that the majority of pollutants are emitted at the surface and then quickly decrease in altitude." has been changed as follows in the revised version to make clear the reason why the emissions are not emitted on the first level only.

In MOCAGE, with the exception of the species emitted from biomass burning [Cussac et al. (2020)], lightning NO$_x$ [Price et al. (1997)] and aircraft [Lamarque et al. (2010)], all of the chemical species sources are injected in the first five levels of the model (up to approximately 500 m). This configuration is necessary for the numerical stability in the lowest model levels. The injection profile implemented follows an exponential decrease from the

surface level of the model (including model orography): $\delta_L = 0.5\delta_{L-1}$, with $\delta_L$ the injection fraction of the mass emitted at the level L of the model. It means that the majority of pollutants are emitted at the surface level and then quickly decrease with altitude. Hereafter, we will refer to "the model surface" when this configuration is used.

L-142/144) *The description of SOA parameterization is very brief, and could be expanded. How well does it perform against organic aerosol observations?*

The parameterization used in MOCAGE is simple. This is why its description is brief. Nevertheless, it was not clear enough. We have improved it in the revised paper.

Secondary organic aerosols are treated in MOCAGE similarly to primary aerosols with its emissions scaled on the primary anthropogenic organic carbon emissions. The scaling factor is derived from aerosol composition measurements [Castro et al. (1999)]. The implementation in MOCAGE was done by Descheemaecker et al. (2019) in the frame of a study on data assimilation for air quality applications.

The evaluation in Descheemaecker et al. (2019) was only done against $PM_{10}$ and $PM_{2.5}$ concentrations over Europe, not targeting specifically the secondary organic aerosols. But note that two general papers describing and extensively evaluating the latest version of the chemistry and aerosols in MOCAGE are in preparation. These papers will include comparisons with observations of different types of aerosols including organic aerosols.

Table 1) *states that the Carn et al. inventory relies on TOMS and OMI, whereas the test mentioned 7 different satellite instruments.*

The mention in the text, that 7 instruments are used in Carn et al (2016), is correct. The mistake in Table 1 has been corrected.

L-253/234) *"One simulation takes into account only anthropogenic emissions": strange, no biomass burning or natural S emissions? → replace by "The first run (NOVOLC) neglects volcanic emissions". Adapt also the rest of the paragraph.*

The sentence has been replaced. We wanted to say that only non-volcanic emissions are injected in this simulation. In the revised version, the general description of the simulations have been improved.

L-291) *"daily mean $SO_2$: satellites do not provide daily means.*

This statement has been corrected in the paper. We wanted to say that we used GOME-2 daily measurements.

L-295) *"thanks to fitting AMF": unclear. As far as I know, the AMF is not fitted.*

We agree that the AMF are not fitted. The sentence was unclear. It is DOAS slant columns which are fitted and then the AMF is applied to produce vertical columns. The description of GOME-2 MetopA dataset is no longer in the revised paper, since we changed our validation strategy.

Table 3) *Table 3 does not bring much, since the global MNMB is given in the text, and the correlation coefficient is negative. Could be moved to the supplement. The global MNMB is not much interesting given the compensation between very high and very low values apparent in Figure 3.*

As explain in the response to major comment 1), the validation strategy has been changed. We do not use anymore the GOME-2 data. Therefore this comments and those (below) regarding lines 360, 361 and 362 are not relevant anymore.

L-360) *"We notice small changes in the vicinity of volcanoes where MNMB score is improved": there are many cases where the MNMB is worsened, including Hawaii and islands (Vanuatu?) in the Southern Pacific.*

L-361) *"FGE is better" → "The FGE is slightly improved"*

L-362) *Some comments are needed concerning the negative value of the correlation coefficient.*

Minor (language) comments:

L-1) *Why "Thus"? The sentence remains true even in the absence of non-linear behaviour.* ⟶ deleted

L-3) *at the global surface → at the global scale (?)* ⟶ corrected

L-4/5) *I would rephrase as "the changes induced by the update of the volcanic emissions inventory are studied using the . . . "* ⟶ rephrased

L-7) *"degassing" → "degassing emissions"* ⟶ rephrased

L-8) *"uncertainties by volcanoes": what does that mean?* ⟶ The sentence was clarified as follows:

Eruptions are provided as daily total amounts of sulfur dioxide (SO$_2$) emitted by volcanoes. Degassing emissions are provided as annual averages with the related mean annual uncertainties of those emissions by volcano.

L-9) *"negligible"* $\longrightarrow$ corrected

L-12) *and elsewhere: remove the dot between Tg and yr* $\longrightarrow$ corrected

L-17) *"necessity of estimates accurate volcanic volcanic sources"* $\rightarrow$ *"need for accurate estimates of volcanic sources"* $\longrightarrow$ rephrased

L-20) *delete "naturel" before "volcanic"* $\longrightarrow$ deleted

L-25) *"Plus"* $\rightarrow$ *"Moreover"* $\longrightarrow$ corrected

L-30) *"to well constrain": ??* $\longrightarrow$ In this sentence, "constrain" means "define". This sentence has been removed because not necessary.

L-43) *"were not very accurate in quantitative, spatial and temporal detection": weird wording, please rephrase* $\longrightarrow$ Unnecessary details were deleted and rephrased as follow:
But at the time these inventories were built, techniques for measuring emission fluxes were not very accurate for the determination of volcanic sources.

L-44) *"used on": ??* $\longrightarrow$ rephrased as "deployed at"

L-46) *Andres and Kasgnoc (1998) cork* $\rightarrow$ *The study of Andres and Kasgnoc (1998)* $\longrightarrow$ replaced

L-51) *"As well": ??* $\longrightarrow$ Unnecessary, removed.

L-55) *"in its work": ??. "more numerous and qualitative data".* $\longrightarrow$ Rephrased as follows:
Carn et al. (2016,2017) sought to compile all those new higher quality data, compared to Andres and Kasgnoc (1998), in order to provide a more representative inventory of volcanic SO$_2$ emissions.

L-59) *"for passive source strength":: ??* $\longrightarrow$ replaced as "for passive emissions"

L-61) *"huge change"* $\rightarrow$ *"stark improvement"* $\longrightarrow$ replaced

L-61) *"last decades studies"* $\rightarrow$ *"studies of the last decades"* $\longrightarrow$ replaced

L-63) *"the radiative forcing induced"* $\rightarrow$ *"the subsequent radiative forcing"* $\longrightarrow$ replaced

L-67) *"on surface species concentration and deposition"* $\rightarrow$ *"on the surface concentration and deposition of sulfur species"* $\longrightarrow$ replaced

L-67) *"We want": ??* $\longrightarrow$ corrected to " We aim"

L-74) *the configuration of simulations with MOCAGE* $\longrightarrow$ rephrased

L-76) *"updating inventory": ??* $\longrightarrow$ mistake, corrected as "updated"

L-76) *"the comparison for"* $\longrightarrow$ replaced

L-77) *"Then"* $\rightarrow$ *"Next"* $\longrightarrow$ replaced

L-83) *"Its use is applied": ??* $\longrightarrow$ cleared up to "It is applied"

L-94) *"the duration emissions"* $\rightarrow$ *"the duration of the emission"* $\longrightarrow$ replaced

L-115) *"from biomass burning process"* $\rightarrow$ *"emitted from biomass burning"* $\longrightarrow$ replaced

L-124) *"completed": ??* $\longrightarrow$ corrected

L-158) *"It was carried out over a period of about 25 years": I suppose you mean the measurements span 25 years. Please rephrase.* $\longrightarrow$ Rephrased as It ranged over a period of about 25 years.

L-167) *"thanks to the similar molecular structure SO$_2$ and ozone": misleading, rephrase or omit; "Thus": ?? The following sentence is unclear. This could be simplified, as not really necessary.*

Indeed, the reference to the similar molecular structure of ozone and sulfur dioxide was too straight forward. The two species have overlapping UV absorption bands (between 300-340 nm). Therefore, TOMS measurement of SO$_2$ is tangled to O$_3$. Krueger et al (1995) explained this phenomenon as follow. "Typically, the amount of sulfur dioxide in the region of the atmosphere that affects TOMS-measured radiances (above the boundary layer) is too small to cause significant absorption. However, a volcanic eruption can produce enough SO$_2$ in a localized region to produce UV absorption comparable to or even exceeding the ozone absorption at the shortest two TOMS wavelengths. In such cases the present TOMS algorithm incorrectly interprets SO$_2$ as enhanced ozone. The problem is to discriminate between sulfur dioxide and ozone.". Thus, an algorithm is needed to discriminate ozone from sulfur dioxide measurements. This level of details is not necessary. The sentence has been deleted.

L-178) *"constancy"* $\rightarrow$ *stability* $\longrightarrow$ replaced

L-178) *"Thus" could be omitted* $\longrightarrow$ deleted

L-178/179) *"in order to incorporate natural variations due to temporal and even chemical inhomogeneities": confusing. Could be omitted.* $\longrightarrow$ deleted

L-181) *"as the one ..."*: replace by *"as being among the largest..."*; *"passive"*: *??* ⟶ replaced and deleted

L-181) *"For them..."* → *"For those volcanoes, fluxes (...) supersede the averages"* ⟶ replaced

L-185) *"Knowing that"* → *"Since"* ⟶ replaced

L-189) *"lowest levels"* or *"lowest level"?* ⟶ As explained in major comment 4), due to numerical issues, it is not possible near the surface to inject emission on a single model level. Therefore, volcanic emissions were previously emitted on the first five levels of MOCAGE, such as anthropogenic and biogenic emissions. It was rephrased in the revised paper.
Since no configuration was developed in MOCAGE to inject volcanic emissions aloft until this study, they were implemented similarly as the other pollution sources. Volcanic $SO_2$ were thus emitted at the model surface (see Sect. **??**). However, the surface elevation of the model (orography) is mainly below the actual elevation of the volcanoes.

L-191) *"technological improvements in satellite technology"*: *awkward* ⟶ clumsy repetition, rephrased as *"With the improvements in satellite technology"*

L-197) *"The work of Carn et al. (...) updates and completes the study of Andres and Kasgnoc (1998).* ⟶ replaced

L-203) *"given is"* → *"given includes"* ⟶ corrected

L-204) *"measured"* → *"estimated"* ⟶ replaced

L-205) *"We will..."*: *is somewhat ambiguous. Within this study or later on?* ⟶ clarified by *"Within this study"*

L-207) *"the daily frequency allows to take into account the eruptions in simulations for the period..."*: *weird statement* ⟶ unnecessary, deleted

L-211) *"could distinguish"* → *"made possible to distinguish"* ⟶ corrected

L-222) *"every day of the year"* → *"throughout the year"* ⟶ replaced

L-236) *"the update of the ..."* → *"the updated"*; *"compiles"* → *"includes"* ⟶ corrected

L-237) *"spread over the globe"*: *??* ⟶ unclear, it means *"worldwide"*, not necessary so deleted

L-244) *Delete words "lat" and "lon"* ⟶ deleted

L-246) *"The same global annual sulfur emissions are computed for all other sources"*: *of course since the same inventories are used !* ⟶ L-245/246 deleted

L-248) *"emissions are"* → *"emissions amount to"*. *You don't need two significant digits after the decimal point, one is enough.* ⟶ replaced

Table 1) *legend "Summary information on"* → *"Summary of"*; *"Nb of volcano"* → *"Numbers of volcanoes"* ⟶ corrected

L-253) *"characteristics"* → *"main features"* ⟶ replaced

L-256) *"However, one injects the volcanic $SO_2$ emissions"* → *"In simulation CARN, volcanic emissions are injected"*. *Adapt also the rest of the sentence.* ⟶ The paragraph was rewritten to make it clearer, as follows:
The first simulation, named REF, takes into account the previous volcanic inventory [from Andres and Kasgnoc (1998)] with the injection at the model surface. The second simulation, named CARNALTI, uses the updated volcanic inventory [from Carn et al. (2016, 2017)] and the new configuration to inject volcanic emissions from the volcano altitude as described in Section 3.2. By comparing REF and CARNALTI runs, we can analyse the changes brought by the updated volcanic emission inventory with respect to the previous one. These two simulations are evaluated in Section 5 and the associated global distribution of sulfur species is compared in Section 6.
In order to distinguish between the impact of the height of emission and of the quantity of $SO_2$ emitted, another simulation, named CARN is run and used for the analysis of the differences between REF and CARNALTI global distribution of sulfur species. Volcanic emissions are from Carn et al. (2016, 2017), like in CARNALTI but they are injected at the model surface, like in REF.

L-261) *"in altitude"* → *"in the vertical"*

L-262) *"Then"*: *?? The entire sentence is weird. You could drop it since you explain what you do in the following sentence.* ⟶ deleted

L-263) *"The CARNALTI run is expected to provide the best..."* ⟶ rephrased

Figure 1) *legend: drop "annual" (since monthly values are shown). "anthropogenic" or "other emissions"?* ⟶ corrected to *"non-volcanic emissions"*.

L-269) *lowest eruptive emission flux (Carn et al., 2016)* ⟶ replaced

L-269) *"is negligible". This sentence could be dropped.* $\longrightarrow$ deleted

L-270) *Why the upper-case AND?; This sentence is weird, not really useful.* $\longrightarrow$ unnecessary, deleted

L-273) *"adds": ??* $\longrightarrow$ This sentence aims to explain that the blue line in Fig. 1 is the addition of non-volcanic emissions (represented is green in Fig. 1) and volcanic emissions from Andres and Kasgnoc (1998). We cleared up the revised paper as follows:

We notice the monthly variation due to non-volcanic emissions (NOVOLC run in green), with less emissions during the northern hemisphere summer and the highest values in the northern hemisphere winter. Volcanic emissions from Andres and Kasgnoc (1998) are stable throughout the year, as we can see in REF run (in blue). They are lower than the volcanic emissions of CARNALTI and CARN runs (in red), with strong constant passive degassing throughout the year and a few sporadically eruptive events.

L-276/277) *"counts": ??; "into": ??, "are"* → *"amount to". Use only one significant digit for the totals* $\longrightarrow$ replaced

L-279) *"current": the use of this word for the previously used inventory is weir. Replace maybe (here and elsewhere) by "previous"* $\longrightarrow$ corrected

L-280) *"inventory against"* → *"not accounted for by". Delete "one"* $\longrightarrow$ corrected

Figure 2) *legend: "round"s* → *"circles"* $\longrightarrow$ corrected

L-284/285) *Weird sentence, provide more direct formulation.* $\longrightarrow$ clarified by "The target chemical species that we evaluate are $SO_2$ and aerosols, since $SO_2$ is the precursor of sulfate aerosols."

L-285) *"benefit". The sentence is true but too obvious.* $\longrightarrow$ deleted

L-289) *"indirectly correlated to $SO_2$* $\longrightarrow$ AOD depends on the quantity of all aerosol species, including sulfate aerosols. And sulfate aerosols are notably formed by sulfur dioxide, therefore $SO_2$ can indirectly impacts the AOD. Nevertheless this statement is unnecessary, deleted

Comments between L-290 to L-306 and between L-356 to L-389 will was ignored since GOME-2 dataset is not considered in the revised paper anymore.

L-299) *"Plus"* → *"In addition"; "presence of offsets"* → *"offsets" (?); "lead"* → *"leads"; "criteria" is plural, replace by "criterion" (if meant as singular)*

L-301) *"subtracted at"* → *"subtracted from"*

L-358) *"higher"* → *"less negative"*

L-359) *"againts"* → *"against"*

Table 4) *"Coorelation"* → *"correlation"; "specifics"* → *"specific"*

L-311) *"low confident"* → *"low-confidence"* $\longrightarrow$ replaced

L-311) *"filtered"* → *filtered out"* $\longrightarrow$ replaced

L-318) *"we can use several statistical metrics": delete, and merge with next sentence "we use the fractional bias..."* $\longrightarrow$ rephrased

L-335/337) *This paragraph could be omitted. Delete "Therefore" from the next paragraph.* $\longrightarrow$ rewritten with a brief description of the new validation strategy.

L-340) *"Plus"* → *"Furthermore". u I don't understand well the rest of the sentence. Rephrase.* $\longrightarrow$ The paragraph has been rewritten and this sentence has been removed.

L-342) *Drop "The" before Zone 1. Same elsewhere.* $\longrightarrow$ deleted

L-343) *You might drop the word "inventory" after the reference. Same remark applies elsewhere in the text.* $\longrightarrow$ deleted

L-346) *"are"* → *"amount to"* $\longrightarrow$ replaced

L-354) *"counting": ??* $\longrightarrow$ replaced by "totalling"

L-698) *the link does not work* $\longrightarrow$ corrected by https://doi.org/10.1016/j.jvolgeores.2016.01.002

Fioletov, V. E., C. A. McLinden, N. Krotkov, K. Yang, D. G. Loyola, P. Valks, N. Theys, M. Van Roozendael, C. R. Nowlan, K. Chance, X. Liu, C. Lee, R. V. Martin, Application of OMI, SCIAMACHY, and GOME-2 satellite $SO_2$ retrievals for detection of large emission sources , JGR Atmospheres, (2013), doi: https://doi.org/10.1002/jgrd.50826.

He, H. and Li, C. and Loughner, C. P. and Li, Z. and Krotkov, N. A. and Yang, K. and Wang, L. and Zheng, Y. and Bao, X. and Zhao, G. and Dickerson, R. R., $SO_2$ over central China: Measurements, numerical simulations and the tropospheric sulfur budget, JGR, (2012), doi : https://doi.org/10.1029/2011JD016473.

Koukouli, M. E. and Balis, D. S. and Theys, N. and Brenot, H. and van Gent, J. and Hendrick, F. and Wang, T. and Valks, P. and Hedelt, P. and Lichtenberg, G. and Richter, A. and Krotkov, N. and Li, C. and van der A, R., OMI/AURA,

SCHIMACHY/ENVISAT and GOME2/MetopA sulphur dioxide estimates; the cas of Eastern Asia, 'ATMOS 2015, Advances in Atmopsheric Science and Application', Heraklion, Greece, June 2015 (ESA SP-375, November 2015)

Rix, M. and Valks, P. and Hao, N. and van Geffen, J. and Clerbaux, C. and Clarisse, L. and Coheur, P-F. and Loyola, D. and Erbertseder, T. and Zimmer, W. and Emmadi, S., Satellite Monitoring of Volcanic Sulfur Dioxide Emissions for Early Warning of Volcanic Hazards, IEEE Journal of Selected Topics in Applied Earth Observations and Remote Sensing, (2009), doi : https://doi.org/10.1109/JSTARS.2009.2031120.

Rix, M. and Valks, P. and Hao, N. and Loyola, D. and Schlager, H. and Huntrieser, H. and Flemming, J. and Koehler, U. and Schumann, U. and Inness, A., Volcanic $SO_2$, BrO and plume height estimations using GOME-2 satellite measurements during the eruption of Eyjafjallajökull in May 2010, JGR, (2012), doi : https://doi.org/10.1029/2011JD016718.

Wang, Y., Wang, J., Tropospheric $SO_2$ and $NO_2$ in 2012–2018: Contrasting views of two sensors (OMI and OMPS) from space, Atmospheric Environment (2020), doi: https://doi.org/10.1016/j.atmosenv.2019.117214.

---

## Author Comment (AC2) · 25 May 2021

Comments on Pasquale Sellitto

We would like to thank Dr Pasquale Sellitto for his comments.

Our response is organised as follows. After each referee's comment (in italic black font) can be found the authors' response (in normal black font), and where needed, the changes made in the manuscript (in blue). In the revised version of the paper, only the significant changes have been coloured in blue to help identifying any new important improvement.

Also to improve the clarity of the paper and following the referees' comments, we have slightly changed the organisation of the paper by splitting section 5 into two. The new Section 5 is only devoted to the evaluation (ex-Sect. 5.1). Section 6 is on the impact of the inventory update on the species concentrations (ex-Sect. 5.2). Also, the purpose of the CARN simulation was not very clear. This simulation is only used to understand the effect of altitude of injection. This is why CARN results are only used now for the analysis of the species concentrations in the new section 6 (ex-Sect. 5.2). The manuscript has been revised accordingly.

Please note that the revised manuscript has been read and corrected by an English native speaker.

The manuscript introduces a new and more detailed volcanic emission inventory (by Simon Carn), input to MOCAGE CTM modelling, and evaluates the improvements brought in the global and regional sulphur budget with respect to older inventories using satellite observation as reference. The topics of this manuscript is of certain interest for multiple communities (atmospheric modellers, atmospheric scientists, climate scientists and volcanologists) and its worth attention. Unfortunately, I have found the following major flaws that, in my opinion, invalidate the results of this work that I think should not be published in the present form.

**Major comments:**

1) In the introduction lines of Sect.3 (L150-155), it is said that  $SO_2$  is the main volcanic effluent and is the only volcanic emission considered in this work. This is absolutely not true. The single most important volcanic effluent is not  $SO_2$  but water vapour, with water vapour/ $SO_2$  emitted mass ratios reaching values as large as a few hundreds. I think that many emissions and near-source volcanic processes can safely be neglected, as a first approximation, like halogen emissions and their impact on sulphate formation, transition metal contribution and other interactions of  $SO_2$ /sulphate with ash including heterogeneous chemistry; nevertheless, volcanic water vapour emissions cannot absolutely be neglected, as well as their in-plume effects on sulphate formation and  $SO_2$  depletion. In my perspective, neglecting water vapour emissions (as said, the dominating gaseous effluent in volcanic degassing) invalidate the results of this work.

There must have been a misunderstanding. We wrote "Among [volcanic gases], sulfur species emitted by volcanoes are mainly sulfur dioxide and hydro-sulfuric acid in much lower quantity". Here, we were not trying to say that sulfur dioxide is the main effluent but to explain that sulfur dioxide is the main effluent for sulfur species. The sentence has been changed in the revised manuscript to make it clearer.

Indeed, water vapour is the most important volcanic effluent and could be taken into account in volcanic emissions. We agree with these statements. There is an effect of volcanic water vapour on the sulfate formation within plumes taking place close in time and space to the emission/eruption.

In our study, we do not take into account this effect as in similar previous studies at the global scale (e.g. the Sheng et al. paper you suggested to cite, updated in Feinberg et al. (2019)). The reasons are that we use a global chemistry-transport model with a resolution of  $1^{\circ} \times 1^{\circ}$  and run over one year. With this resolution, if we were to include water vapour volcanic emissions and associated sulfate production, the water vapour emitted would be diluted into the grid box (about 100km  $\times$  100km) such that its effect on the acceleration of sulfate formation would be negligible. More importantly, the effect of the water emitted by the volcances is mainly important in the first stages of the plumes in the vicinity of volcances, small scale processes that we cannot and do not intend to represent in our global model at  $1^{\circ} \times 1^{\circ}$  resolution. The aim of the paper is not to focus on small time and space scales close to the volcanic emissions but to assess the impact at the global scale of sulfur emissions of all volcances on the sulfur budget. Note also that MOCAGE simulations use meteorological analyses using data assimilation of water vapour information. Therefore, some of the volcanic water vapour can possibly be taken into account in the meteorological analyses via data assimilation of water-related observations.

2) I agree with Referee #1 on the fact that ACSAF GOME-2 retrievals are not a good choice for the validation of the MOCAGE simulations. I'd also mention that, differently to what said at L287, GOME-2 data are not completely independent on the OMI and TOMS input data to your inventories: the instruments operate in the UV spectral range and use similar spectral ranges and  $SO_2$  absorption structures for the retrieval. Why not using infrared instruments as IASI, instead?

We agree with the reviews that GOME-2 MetopA  $SO_2$  dataset shows unrealistic features in some regions. Although we had applied filtering on these data, it was not enough to remove all the noise and artefacts.

Concerning IR instruments, such as IASI, they have the advantage to be fully independent of OMI since using different wavelengths. Moreover they have the ability to retrieve  $SO_2$  columns at high latitudes in winter or at night. However, IASI is mainly sensitive to  $SO_2$  in the mid and upper troposphere but not very much in the planetary boundary layer (or under 5 km). Even with more sophisticated algorithms designed to extract information below 5 km, the estimates of  $SO_2$  columns are shown to be less accurate in the lower troposphere and to underestimate small emissions sources [Carboni et al (2012), Taylor et al (2018)]. This is why in the literature, IASI was mainly used to study eruptive events, emitting at higher altitudes [Clarisse et al (2008, 2014), Carboni et al (2016, 2019)]. Therefore, we think that IASI measurements are not suitable for the model evaluation for the year 2013, since 2013 has very few eruptive events and thus volcanic emissions are mostly emitted below 5km. Still, it would be interesting to use IASI  $SO_2$  columns for a year with more and higher eruptions.

Concerning the validation strategy in the paper, we no longer used GOME-2 SO2 columns but OMI SO2 columns as suggested by referee 1. Even if OMI has been used for in the Carn's inventory, it has the finest resolution and it is the most accurate instrument in 2013 to retrieve SO2 total columns over passively emitted volcanoes which altitudes are generally around 2-3 km. We also changed the approach chosen for the statistical evaluation based on the analysis of the literature. In satellite derived products, the relative uncertainties on SO2 columns are large where the SO2 signal is low, in particular for background SO2 conditions. This is why in the literature, the SO2 satellite comparisons or model evaluations focus on specific areas close to SO2 sources [*e.g.* He et al. (2012), Fioletov et al. (2013), Wang and Wang (2020)]. Similarly to these studies, our new strategy is to perform the model evaluation only in the vicinity of the volcanic sources. The comparison between the model and OMI SO2 columns clearly show an improvement of the model performances in CARNALTI simulation. Section 4.2 "Observations used for the evaluation strategy and associated results. Further details are explained in the answer to Referee #1.

3) Also, the choice of MODIS AOD is quite debatable strategy. MODIS AOD is linked to all aerosols, not only sulphates. How do you separate sulphate aerosols from the other aerosol typologies/composition/sources? For example, in "Region 3 (Mediterranean)" dust is, on average, overwhelmingly dominant with respect to sulphate aerosols. How can you check the improvement of volcanic  $SO_2$  sources in such an environment, due to the expected small sulphate signal?

We are aware that MODIS AODs include all aerosols but satellite observations of sulfate aerosols only at the global scale are not available. MODIS AOD is an alternative allowing us to make an indirect evaluation since it takes into account all types of aerosols, sulfate included. Between the different simulations, only  $SO_2$  volcanic emissions are modified. So, the changes between the model simulations with respect to MODIS AODs come only from sulfate aerosols.

Regarding dusts, they contribute largely to AODs seasonally in some regions and can partially/totally hide the sulfate contribution to AODs. Still note that dusts also help forming sulfate aerosols. In the paper, we had highlighted that in some regions (mostly polluted areas), those changes are very small because hidden by anthropogenic emissions. This is further discussed in the revised version.

Note also that in the validation process, with MODIS or another instrument, we are interested in the differences between REF and CARNALTI simulations. In general, the validation of the AODs show a decreasing FGE and an increasing correlation between REF and CARNALTI. We consider this result satisfying.

4) In addition, how to interpret the results of the comparisons in Tables 3 and 4? Am I wrong to say that observations and simulations compare very weakly? This is also the case if looking at the (necessary in the main text) Figure S1. The simulations and the observations seem to not describing the same  $SO_2$  fields. Results for aerosols compare better but, in my opinion, only because the aerosol fields are dominated by other aerosols (and MODIS is more sensitive to higher altitudes aerosols than boundary layer aerosols, so again probably large dust plume lofted by convection).

Tables 3 and 4 are results from the comparison with GOME-2 Metop A. Since we now use OMI instead of GOME-2, the part of the paper concerning the  $SO_2$  evaluation has been fully revised (see answer to Major Comment 2).

Concerning aerosols validation, Fig. 11b clearly shows that volcanic emissions can have a strong contribution in aerosol sulfate concentrations. Even over polluted area, such as the Mediterranean Sea, sulfate aerosols from volcanic emissions contribute to about 10-20%. This is not negligible.

5) In general, the manuscript is poorly written and needs a thorough linguistic review. The description of the scientific context is quite approximative and a lot of key references are lacking – please see specific comments in the following.

The manuscript has been revised and corrected following the suggestion of all reviewers and fully checked by a native English speaker. Several references have been added in the revised manuscript. We preferentially chose those focusing on global studies rather than case studies, the latter being of lower relevance for the aims of the paper.

Technical comments:

1) L1: "The contribution...": what contribution?

Here, we talk about the contribution of volcanic sulfur emissions, with respect to other sulfur emissions. We emphasise that volcanic emissions allow more sulfur species to remain into the troposphere, and especially sulfate aerosols. But this sentence was not appropriate here  $\rightarrow$  deleted.

2) L9: "negliable"  $\rightarrow$  "negligible"  $\rightarrow$  corrected

3) L20: Here and all the following discussions (including comparisons with your results): recent assessments of sulphur budget should be discussed here, like (for volcanically quiescent conditions, so of large interest for our study): Sheng et al (2015).

Thank you for the suggestion of the Sheng et al. (2015)'s reference which was updated with improved model simulations in Feinberg et al. (2019). Even if these two studies focus on the stratosphere, they provide a sulfur budget estimate also for the troposphere. Comparisons to Sheng et al. (2015) and Feinberg et al. (2019) have been included in the revised manuscript.

4) L-20: Also, for climate impacts, this should be cited and possibly discussed: Kremser et al. (2016) We have included the reference as suggested but no addition discussion since the climate impact of sulfur is out of the scope of our paper.

5) L22: "variation of climate": You mean "climate forcing"? (in this case, please specify that you're not talking about  $SO_2$  but sulphate aerosols)  $\longrightarrow$  We meant sulfate aerosols climate forcing. This has been corrected.

6) L23-24: "SO2 emissions had become a major concern in environmental policies, leading to strong reductions in anthropogenic emissions in recent decades": Not everywhere. Please differentiate geographically between decreasing, stationary and increasing emissions regions and add a reference.

Indeed, changes in anthropogenic  $SO_2$  emissions are not similar worldwide. Details were added in the revised manuscript as follows:

In some regions of the world, these policies led to strong reductions in anthropogenic  $SO_2$  emissions in the recent decades [Fioletov et al. (2016), Krotkov et al. (2016), Aas et al. (2019)]. Over North America and Europe, emissions strongly decreased between 2005 and 2015. In the East Asian region, the decrease only happened after 2010 [Sun et al. (2018)]. On the contrary, over India, emissions strongly increased. And over other large  $SO_2$ -emitting regions (Mexico, South Africa, Russia or Middle East), they remained stable since 2000. However, the decrease in anthropogenic  $SO_2$  emissions over Europe and North America was sufficient to induce an overall decrease at the global scale.

7) L23-24: "Thus, the relative proportion of volcanoes in the total sulfur emission sources tends to increase.": Due to different regional trends of anthropogenic emissions, this statement sounds just arbitrary (unless you have specific references that I don't know).

Despite the increase of anthropogenic SO2 emissions in India, the statement about the decrease in anthropogenic SO2 emissions at the global scale is true. With respect to anthropogenic emissions, we think that it is acceptable to say that at the global scale, the relative proportion of volcanic emissions in the total sulfur emission sources tends to increase.

8) L26-27: "is greater in altitude": you might mean: "increases with altitude"  $\rightarrow$  corrected

9) L27-28: "Thus, we now... emissions": not clear, please rephrase  $\longrightarrow$ . Removed

10) L27: "longer": "for longer time periods"?  $\longrightarrow$  corrected

11) L29: "these variations": which variations?  $\longrightarrow$  "these" unnecessary and deleted to clarify.

12) L44-45: Please change the phrasing here: there are very few "easy-to-access" volcanoes (Masaya can be mentioned, maybe), while normally the internal processes themselves build "uneasy-to-access" morphological structures for volcanoes.  $\rightarrow$  rephrased

13) L51: "information on injection altitude is available". The information on the altitude is still very limited. These are observing systems that have a few units of Degrees of Freedom in vertical profile observations of  $SO_2$ , mostly between 1 and 2.0-2.5, so not allowing for detailed altitude information. Please mention this in the text and smooth this statement.

We agree that emissions inventory, and especially those built by satellite observations, should not be considered as the absolute truth, because there are uncertainties in the retrievals, both for the  $SO_2$  quantities and the plume height. Moreover, in MOCAGE, there are only 47 vertical levels. Into the free troposphere, each levels are separated by at least 200-300 m and volcanic emissions are injected on the closest vertical level of the plume altitude. Therefore, this adds more uncertainties to the plume height. We added a statement in Section 3.2 "New volcanic sulfur inventory" as follows:

Note that depending on the instrument used, the retrieval of the plume altitude can differ. Therefore, there are uncertainties on the altitude information provided by this inventory.

14) L56: what do you mean with "more numerous and qualitative data"?

This sentence means that, with the improvement in the retrieval of  $SO_2$  emissions by satellites, Carn et al (2016,2017) inventory includes more data over more volcanoes and with a higher quality, with respect to Andres & Kasgnoc (1998) inventory  $\longrightarrow$  rephrased as follows:

Carn et al (2016,2017) sought to compile all those new higher quality data, compared to Andres & Kasgnoc (1998), in order to provide a more representative inventory of volcanic  $SO_2$  emissions.

15) L63-65: "In contrast, few studies focus on the impact on tropospheric composition including air quality, with the exception of case studies of volcanic eruptions...": This is not true. Please look at the following papers of my research group, that aimed at the impact of volcanic activity, including passive degassing of selected volcanoes, on the tropospheric composition and air quality. Please correct the wrong statement and cite the previous work mentioned above.

There is a misunderstanding here. In this sentence, "In contrast" was a reference to studies at the global scale only, similar to ours. Some studies, such as yours, analysed the impact of volcanic degassing (and not only eruptions), but at the regional scale. Thank you for the references. We have added more prescise information in the revised version.

At the global scale, numerous studies aim at the assessment of the dispersion of sulfate aerosols and the subsequent radiative forcing [Graf et al. (1997,1998), Gasso et al. (2008), Ge et al. (2016)]. In contrast, regarding their impact on tropospheric composition, including air quality, several case studies at the regional scale have been analysed [e.g. Colette et al. (2010), Schmidt et al. (2015), Boichu et al. (2016,2019), Sellitto et al. (2017)] but very few studies at the global scale have been conducted [Chin et al. (1996), Sheng et al. (2015), Feinberg et al. (2019].

16) Section 2.5: What about the vertical transport, which can pose problems for the modelling of confined plumes, like volcanic plumes, and is discussed in the following paper [Lachantre et al. (2020)]?

Lachatre et al (2020) is a good reference about the vertical diffusion of plumes linked to the modeling of vertical transport. But in our analysis at the global scale, we do not focus on the study of individual volcanic plumes but on the global fate of the sulfur volcanic emissions and in particular on their impact once dispersed. As in all global models, MOCAGE description of volcanic plumes is limited by both the vertical (at least 200-300m in the free troposphere) and the horizontal  $(1^{\circ} \times 1^{\circ})$  resolutions not allowing the detailed modeling of individual plumes.

18) L165: "calm": What do you mean with "calm"? "Non-eruptive"?

"Calm eruptive conditions" do not mean "non-eruptive", but with small eruptions and not strong eruptions. The term used in Andres & Kasgnoc (1998) is "quiet"  $\rightarrow$  corrected

19) L167: the reference to molecular structure sounds strange here. You might want to say that "SO2 and ozone have absorption bands at overlapping spectral regions" (which is linked to molecular structure) or something like.

The reference to the similar molecular structure of ozone and sulfur dioxide was difficult to understand. The two species have overlapping UV absorption bands (between 300-340 nm). Therefore, TOMS' measurement of  $SO_2$  is tangled to  $O_3$ . Krueger et al (1995) explained this phenomenon as follow. "Typically, the amount of sulfur dioxide in the region of the atmosphere that affects TOMS-measured radiances (above the boundary

layer) is too small to cause significant absorption. However, a volcanic eruption can produce enough SO2 in a localized region to produce UV absorption comparable to or even exceeding the ozone absorption at the shortest two TOMS wavelengths. In such cases the present TOMS algorithm incorrectly interprets SO2 as enhanced ozone. The problem is to discriminate between sulfur dioxide and ozone.". Thus, an algorithm is needed to discriminate ozone from sulfur dioxide measurements. This level of details is not necessary. The sentence has been deleted.

20) L181: "... as one of the largest passive emitters": Clumsy phrasing. Please rephrase.  $\longrightarrow$  rephrased as follows:

Etna in Sicily, Kilauea and the Kilauea Rift Zone in Hawaii, which are known as being among the largest emitters of  $SO_2$ .

21) L181-182: Please add details on the sources of these flux information.

The original statement in Andres & Kasgnoc (1998) is :"For three sites, however, personal communications supplanted the average. These personal communications relied upon published and unpublished data for Etna, Kilauea and Kilauea East Rift Zone." No other information is available. Therefore, we cannot add details in the paper.

22) L183-184: This is very unclear. Please clarify.

We agree that the explanation we gave in the paper on sporadic eruptions can be confusing. We could have explained it as follows: "With regard to sporadic eruptions that are considered for 25 volcanoes, Andres & Kasgnoc (1998) use the maximum flux reported during the period and assume an average of 7 eruptions per year, each lasting one day. From this, sporadic eruptions account for less than 1 % of the total annual emissions in their inventory". However, in the next paragraph, we explain that eruptions are not taken into account in the model. Therefore, this level of details is not needed  $\rightarrow$  The sentence has been deleted.

23) L188-189: You mean that volcanic  $SO_2$  is emitted at the surface (including orography)? Is orography "smoothed" by the average in-grid topology? This aspect is very important e.g. for Etna. Even in case of passive degassing, its emissions are released at, at least, 3000 m altitude and episodic eruptions can reach, for Etna and Kilauea, quite higher altitudes.

The model surface altitude corresponds to the model orography which is calculated as the average in-grid topography. This means that in the previous version of MOCAGE, the volcanic emissions which are emitted at the surface are mostly under the actual volcano altitude. This is now clearly mentioned in the revised paper.

24) Section 3.2: there are many repetitions. In general, all the paper should be condensed and repetitions should be suppressed.  $\rightarrow$  corrected

Section 3.2 has been changed according to referees' comments and overall reduced in the revised version. 25) L224-226: "We implemented...emissions": Why this parameterisation is not described in details here? How it compares to established parameterisations like the one of Mastin et al (2014)?

For each eruption, we use the altitude of the volcano and the height of the eruption given in Carn et al. (2016) inventory. This information is derived from the analysis of nadir UV and IR satellite observations. Therefore, we do not need to make an estimation of the eruption height by the use of a parameterizations like the one proposed by Mastin et al. (2014). Still, in the model, we have to distribute vertically the mass of SO2 given in the inventory. In MOCAGE, we distribute the eruption emission mass from the model level of the volcano altitude to the model level of the plume top height, following an "umbrella" profile similar to that used in other models (Freitas et al. 2011 in CCATT-BRAMS and Stuefer et al. 2013 in WRF-Chem). In practice, the plume follows an almost linear profile with increasing altitude from the volcano vent and then opens into a parabola containing 75 % of the gases in mass into the top third of the plume. This paragraph has been re-written in the revised version in order to be clearer.

26) L239: "Finally, the availability of emission heights in this inventory gives a better description of the emission". At this point I think it is necessary to discuss the limitations in the vertical characterisation of volcanic emissions in the new inventory and the satellite observations used to build it, so to not oversell your new simulations.

We fully agree and we have added a sentence on the uncertainties of the inventory and satellite observations in this section. We also added a paragraph on this subject in the conclusion.

27) Figure 1 and most figures: Please use larger text and labels.  $\rightarrow$  corrected

28) L269: "lowest eruptive. . .negligible in 2013": How much this is "low"? Is it really negligible? How do you qualify this as "negligible"?

This information is given just after "the total 2013 annual emissions in Carn et al (2016, 2017) inventory amount to 23.7 Tg of SO2 (or 11.8 Tg S), with 23.5 Tg of passive degassing SO2 and 0.2 Tg of eruptive emission (< 1 % of the total amount of volcanic SO2 emission)". This part of the paragraph was changed in order to make this clearer.

29) L272: reference to summer and winter: Please correct to "northern hemisphere summer/winter" and adapt the discussion.  $\rightarrow$  rephrased.

30) L284-285: "Due" and "since" in the same sentence is quite clumsy. Please rephrase.  $\longrightarrow$  rephrased as follows:

The target chemical species that we evaluate are  $SO_2$  and aerosols, since  $SO_2$  is the precursor of sulfate aerosols.

31) Section 4: see Major Comments  $2-3 \longrightarrow$  Taken into account. This section has been revised (see answers to comments 2-3)

32) Section 4.2.2 title: "MODIS Aerosol Optical Depth"  $\rightarrow$  corrected

33) L349: Please check altitude of Mount Etna, this is not the right altitude.

We agree that the altitude of Mount Etna is about 3330 m but the altitude provided in Carn et al. (2017) inventory is 2711 m. No information is given in the documentation/publication why this altitude is lower in the inventory. It possibly accounts for passive emissions from volcano flanks or a mistake. Nevertheless, the aim of the paper is to implement a new volcanic emissions inventory and to evaluate it as a whole in the model, even if there are possible uncertainties in the altitude of the volcanoes or other parameters.

34) Section 5: see Major Comment 4  $\longrightarrow$  taken into account

35) Section 5: It looks like some of the Figures in the Supplements are needed here in the main text, e.g. S1  $\rightarrow$  with the new validation strategy, now not necessary, deleted.

36) L430: "(industries...': and dust, of course  $\rightarrow$  added

37) Figure 8: This figure would be largely more useful with an altitude vertical axis (instead of pressure).  $\rightarrow$  We have added the altitude axis and enhanced the labels for the pressure axis.

38) L517: "This corresponds...eruption": This is quite straightforward interpretation of these results, but it is important to stress the fact that 2013 is not a "normal" year as even a small number of explosive volcanic eruptions can change the vertical distributions of Figure 8 at the global scale. This has to be discussed and the limits of your simulation (a" predominantly passive degassing" year) must be clearly stated.

There is a misunderstanding here because this sentence is a comment on Graf et al (1997) results. We changed the paragraph as follows:

For volcanic sulfate, the maximum is between 850 and 450 hPa but four times smaller than for other sources and without any specific peak associated to passive degassing or eruptive emissions. These results are different from Graf et al. (1997), which shows that the vertical distribution of volcanic sulfate aerosols is comparable to anthropogenic and biomass burning sulfate and is even dominant between 800 and 300 hPa (the altitude of volcanic emissions, mainly from eruption). This difference between our study and Graf et al. (1997) can be explained by the quantity of  $SO_2$  emitted by eruptions. In 2013, only a few eruptive events occurred while almost 30% of volcanic emissions in Graf et al. (1997) are eruptive. Therefore, with a greater amount of volcanic emissions injected at higher altitude in Graf et al. (1997), the potential to form sulfate aerosols is greater than in our study. This can explain the greater efficiency of 2.63 in the tropospheric sulfate burden in Graf et al. (1997) compared to 1.89 in our study.

In Fig. 8, we had not clearly discussed about the impact of eruptive emissions on the vertical distribution. We added this statement in the revised paper:

[SO2 vertical profile] There is no contribution below 950 hPa but there are three maxima above; one at 850 hPa (about 1500 m) due mostly to passive degassing, another around 680 hPa (about 3300 m) due to passive degassing from high-altitude volcanoes and eruptions, and the last one around 450 hPa (about 6000 m) due to high-altitude eruptions. It is noteworthy that even with few eruptive events during the year 2013, the volcanic SO2 vertical distribution is affected by them.

Carboni, E. and Grainger, R. and Walker, J. and Dudhia, A. and Siddans, R., A new scheme for sulphur dioxie retrieval from IASI measurements: application to the Eyjafjallajökull eruption of April and May 2010, Atmospheric Chemistry and Physics, (2012), doi : https://doi.org/10.5194/acp-12-11417-2012.

Carboni, E. and Grainger, R. and Mather, T. A. and Pyle, D. M. and Thomas, G. E. and Siddans, R. and Smith, A. and Dudhia, A. and Koukouli, M. E. and Balis, D., The vertical distribution of volcanic SO2 plumes measured by IASI,

Atmospheric Chemistry and Physics, (2016), doi: https://doi/org/10.5194/acp-16-4343-2016.

Carboni, E. and Mather, T. A. and Schmidt, A. and Grainger, R. and Pfeffer, M. A. and Ialongo, I. and Theys, N., Satellited-derived sulfur dioxide (SO2) emissions from the 2014-2015 Holuhraun eruption (Iceland), Atmospheric Chemistry and Physics, (2019), doi: https://doi/org/10.5194/acp-19-4851-2019.

Clarisse, L. and Coheur, P.F. and Prata, A.J. and Hurtmans, D. and Razavi, A. and Phulpin, T. and Hadji-Lazaro, J. and Clerbaux, C., Tracking and quantifying volcanic SO2 with IASI, the September 2007 eruption at Jebel at Tair, Atmospheric Chemistry and Physics, (2008), doi: https://doi.org/10.5194/acp-8-7723-2008.

Feinberg, A. and Sukhodolov, T. and Luo, B-P. and Rozanov, E. and Winkel, L. H. E. and Peter, T. and Stenke; A., Improved tropospheric and stratospheric sulfur cycle in the aerosol–chemistry–climate model SOCOL-AERv2, Geoscientific Model Development, (2019), doi: https://doi.org/10.5194/gmd-12-3863-2019.

Fioletov, V. E., C. A. McLinden, N. Krotkov, K. Yang, D. G. Loyola, P. Valks, N. Theys, M. Van Roozendael, C. R. Nowlan, K. Chance, X. Liu, C. Lee, R. V. Martin, Application of OMI, SCIAMACHY, and GOME-2 satellite SO2 retrievals for detection of large emission sources, JGR Atmospheres, (2013), doi: https://doi.org/10.1002/jgrd.50826.

He, H. and Li, C. and Loughner, C. P. and Li, Z. and Krotkov, N. A. and Yang, K. and Wang, L. and Zheng, Y. and Bao, X. and Zhao, G. and Dickerson, R. R., SO2 over central China: Measurements, numerical simulations and the tropospheric sulfur budget, JGR, (2012), doi: https://doi.org/10.1029/2011JD016473.

Koukouli, M. E. and Balis, D. S. and Theys, N. and Brenot, H. and van Gent, J. and Hendrick, F. and Wang, T. and Valks, P. and Hedelt, P. and Lichtenberg, G. and Richter, A. and Krotkov, N. and Li, C. and van der A, R., OMI/AURA, SCHIMACHY/ENVISAT and GOME2/MetopA sulphur dioxide estimates; the cas of Eastern Asia, 'ATMOS 2015, Advances in Atmospheric Science and Application', Heraklion, Greece, June 2015 (ESA SP-375, November 2015)

Rix, M. and Valks, P. and Hao, N. and van Geffen, J. and Clerbaux, C. and Clarisse, L. and Coheur, P-F. and Loyola, D. and Erbertseder, T. and Zimmer, W. and Emmadi, S., Satellite Monitoring of Volcanic Sulfur Dioxide Emissions for Early Warning of Volcanic Hazards, IEEE Journal of Selected Topics in Applied Earth Observations and Remote Sensing, (2009), doi: https://doi/org/10.1109/JSTARS.2009.2031120.

Rix, M. and Valks, P. and Hao, N. and Loyola, D. and Schlager, H. and Huntrieser, H. and Flemming, J. and Koehler, U. and Schumann, U. and Inness, A., Volcanic SO2, BrO and plume height estimations using GOME-2 satellite measurements during the eruption of Eyjafjallajökull in May 2010, JGR, (2012), doi: https://doi.org/10.1029/2011JD016718.

Sheng, J-X. and Weisenstein, D. K. and Luo, B. P. and Rozanov, E. and Stenke, A. and Anet, J. and Bingemer, H. and Peter, T., Global atmospheric sulfur budget under volcanically quiescent conditions: Aerosol-chemistry-climate model predictions and validation, Journal of Geophysical Research: Atmospheres, (2015), doi: https://doi.org/10.1002/2014JD021985.

Taylor, I. and Preston, J. and Carboni, E. and Mather, T. A. and Grainger, R. G. and Theys, N. and Hidalgo, S. and McComick Kilbride, B., Exploring the Utility of IASI for Monitoring Volcanic  $SO_2$  Emissions, Journal of Geophysical Research: Atmospheres, (2018), doi: https://doi.org/10.1002/2017JD027109.

Wang, Y., Wang, J., Tropospheric SO2 and NO2 in 2012–2018: Contrasting views of two sensors (OMI and OMPS) from space, Atmospheric Environment (2020), doi: https://doi.org/10.1016/j.atmosenv.2019.117214.

---

## Author Comment (AC3)

Comments on Anonymous Referee #2

We would like to thank the Anonymous Referee #2 for its comments that helped improving the paper. Our response is organised as follows. After each referee's comment (in italic black font) can be found the authors' response (in normal black font), and where needed, the changes made in the manuscript (in blue). In the revised version of the paper, only the significant changes have been coloured in blue to help identifying any new important improvement.

Also to improve the clarity of the paper and following the referees' comments, we have slightly changed the organisation of the paper by splitting section 5 into two. The new Section 5 is only devoted to the evaluation (ex-Sect. 5.1). Section 6 is on the impact of the inventory update on the species concentrations (ex-Sect. 5.2). Also, the purpose of the CARN simulation was not very clear. This simulation is only used to understand the effect of altitude of injection. This is why CARN results are only used now for the analysis of the species concentrations in the new section 6 (ex-Sect. 5.2). The manuscript has been revised accordingly.

Please note that the revised manuscript has been read and corrected by an English native speaker and that we have added co-authors to the paper that contributed to the responses to the referees and to the revised version.

The paper by Lamotte et al. studies the effect of  $SO_2$  volcanic degassing emissions on the tropospheric sulfur budget, using the MOCAGE global CTM implemented with new volcanic emission inventory of Carn et al. (2016,2017). By model sensitivity tests, tropospheric  $SO_2$ , sulfate and AODs simulated with new and old emission inventories for the year 2013 are compared and validated against  $SO_2$  GOME-2 and AOD MODIS satellite data sets. The results show that the new inventory (CARNALTI) is the best with reference to satellite observations. Such kind of study is interesting and should provide important information for understanding the sulfur global budget. On the other hand, in the opinion of this referee, the manuscript still needs improving a lot before acceptance for publication.

**1A) I agree with Referee #1 and Dr. Pasquale Sellitto that other observational data should be used for the validation of the model simulations.**

We agree with the reviewer that the Metop-A GOME-2 SO2 columns presented show unrealistic features in some regions. Not being experts on satellite observations, we had chosen for the model evaluation to use GOME-2 MetopA SO2 columns from DLR provided by ACSAF (ex- O3F-SAF) because those data provide an independent measurement of SO2 with respect to OMI (used in the volcanic emission inventory). Indeed, these data present artefacts and noise. Although we had applied filtering, this was not enough to remove all the unrealistic features. This is probably the reason why these data were mainly used in the literature not at the global scale but on case studies at the regional and local scales [Rix et al (2009,2012), Koukouli et al (2015)], and to detect very large emission sources [Fioletov et al (2013)]. Note that we also investigated the use of GOME-2 MetopB SO2 columns from DLR by ACSAF (ex- O3F-SAF) but the results showed similar unrealistic features in some regions as in GOME-2 MetopB SO2 columns. Concerning IR instruments, such as IASI, they are mainly sensitive above 5km which is too high for our study focused on passive emissions.

This has lead us to change our evaluation strategy. As suggested by Referee #1, we choose in the revised version to use OMI SO2 columns data for the model evaluation. We also changed the approach chosen for the statistical evaluation based on the analysis of the literature. As for all satellite derived products, the relative uncertainties on SO2 columns are large where the SO2 signal is low, in particular for background SO2 conditions. This is why in the literature, the SO2 satellite comparisons or model evaluations focus on specific areas close to SO2 sources [*e.g.* He et al. (2012), Fioletov et al. (2013), Wang and Wang (2020)]. Similarly to these studies, our new strategy is to perform the model evaluation only in the vicinity of the volcanic sources. For each volcano, we select 9 model grid points (representing a square of  $3^{\circ}$  longitude x  $3^{\circ}$  latitude) with the middle point being where the volcano is located. The comparison between the model and OMI SO2 columns clearly show an improvement of the model performances in CARNALTI simulation. Section 4.2 "Observations used for the evaluation of the simulations" and 5.1 "Evaluation of the simulations" were rewritten to explain our new model evaluation strategy and associated results. Further details are explained in the answer to Referee #1.

1B) In addition, I am very concerned about the effects of the volcanic emission heights on the simulated results.

Tables 1 and 2 present different simulation scenarios, but I missed the detailed information on emission heights of the new inventory of Carn et al. (2016,2017). As I understand, the emission heights are provided in the work of Carn et al. (2016,2017), but for the sensitivity test, all volcanic emissions (eruptive and passive) were arbitrarily forced to the model surface for the scenario CARN.

We were not clear enough, but your understanding was right. The aim of the CARN simulation is to make a sensitivity run in support of the analysis of the differences between REF and CARNALTI. The major differences between Andres and Kasgnoc (1998) and Carn et al (2016,2017) are the updated quantities of volcanic emissions and the information on plume altitude. CARN simulation allows us to distinguish between the impact of the height of emission and of the quantity of SO2 emitted. This is why in the revised manuscript, CARN results are only used for the analysis of the species concentrations in the new section 6 (ex-Sect. 5.2) The paragraph describing the different simulations was rewritten as follows:

The first simulation, named REF, takes into account the previous volcanic inventory [from Andres and Kasgnoc (1998)] with the injection at the model surface. The second simulation, named CARNALTI, uses the updated volcanic inventory [from Carn et al. (2016, 2017)] and the new configuration to inject volcanic emissions from the volcano altitude as described in Section 3.2. By comparing REF and CARNALTI runs, we can analyse the changes brought by the the updated volcanic emission inventory with respect to the previous one. These two simulations are evaluated in Section 5 and the associated global distribution of sulfur species is compared in Section 6.

In order to distinguish between the impact of the height of emission and of the quantity of  $SO_2$  emitted, another simulation, named CARN is run and used for the analysis of the differences between REF and CARNALTI global distribution of sulfur species. Volcanic emissions are from Carn et al. (2016, 2017), like in CARNALTI but they are injected at the model surface, like in REF.

**1C) While the $SO_2$ tropospheric columns are compared, can the vertical distributions of $SO_2$ , sulfate or aerosols be better simulated using CARNALTI than with CARN?**

With different altitudes of emission between CARN and CARNALTI simulations, we expect higher aerosol content at altitude with CARNALTI simulation (as explained in section 5.2 "By injecting volcanic emission in altitude with the new configuration in the simulation CARNALTI, less sulfur species remain at the surface and therefore aerosols are spread further from the volcanoes (see Fig 7b)"). Moreover, the vertical variability of winds also induces differences in the horizontal aerosol distribution. In CARNALTI, SO2 volcanic emissions are more realistically distributed vertically leading to an expected improvement of the overall vertical and horizontal distributions of  $SO_2$ .

**2) 2.3 Emissions (L 109-114): The emission inventories (MACCity and GFAS) are for the years before 2010, earlier than the simulated year 2013. Can this affect the comparing results?**

As written in the paper, anthropogenic emissions from MACCity inventory and biogenic emissions from MEGAN-MACC inventory are representative of the year 2010. However, the differences between 2010 and 2013 emissions are not very important (see Figure 1). At the global scale,  $SO_2$  emissions are only about 1% higher than in 2013. Locally, it represents only a reduction of 8% over oceans/seas between 2010 and 2013, and an increase of 7% over North Africa. Therefore, the expected impact of the use of the 2010 emissions instead of 2013 is low. We have added a sentence in the revised manuscript stating that the differences between 2013 and 2010 are small.

For GFAS products, it is a database available since 2012 with daily biomass burning emissions for each year since then. In this study, we used the daily GFAS data for 2013. We make it clearer in the revised version of the paper.

**3A) 2.4.1 Gaseous species (L 123-125): Two schemes, RACM and REPROBUS, are used for tropospheric chemistry and stratospheric chemistry, respectively. How the model grid cells are distinguished between the troposphere and the stratosphere so that only one of them is applied?**

The chemical scheme used in MOCAGE is a merge of RACM and REPROBUS so that no distinction between the troposhere and the stratosphere is needed. This means that all chemical species are defined at all gridpoints in MOCAGE. The following changes are made in the manuscript.

The MOCAGE chemical scheme is named RACMOBUS. It merges two chemical schemes representing the tropospheric and stratospheric chemistry. The first one, the Regional Atmospheric Chemistry Mechanism (RACM) (Stockwell et al. 1997), completed with the sulfur cycle [details in Guth et al. (2016)], represents tropospheric species and reactions. The second one, REactive Processes Ruling the Ozone BUdget in the Stratosphere (REPROBUS), provides the additional chemistry species and reactions relevant for the stratosphere, in particular long-lived ozone depleting substances (Lefevre et al. 1994).

Figure 1:  $SO_2$  total emissions (all sectors) from MACCity inventory for the year (upper-left) 2010 and (upper-right) 2013. (lower-left) Relative difference between 2010 and 2013  $SO_2$  emissions in MACCity. (lover-right) Time series of the total annual  $SO_2$  emission in MACCity inventory from 2010 to 2020.

**3B) How are stratospheric tracers (e.g., CFCs and OCS) and tropospheric tracers (e.g., NMVOCs) treated in the model grid cells?**

Long lived species relevant for the stratosphere (e.g., CFCs) are fixed at the surface, similarly to many other global models. Tropospheric VOCs undergo chemical processing in the troposphere leading to negligible concentrations reaching the stratosphere as expected. Still, there is one exception of species that is represented with two different model variables. It is for  $H_2SO_4$  for which a climatology is used in the stratosphere and which is treated as a 'normal' species in the troposphere In the paper, we analyse tropospheric sulfur only.

**3C) Can TUV calculate the photodissociation rates of stratospheric chemical tracers?**

Yes, TUV model can calculate photodissociation rates for both the troposphere and stratosphere. This detail is now explained in the revised manuscript.

Fioletov, V. E., C. A. McLinden, N. Krotkov, K. Yang, D. G. Loyola, P. Valks, N. Theys, M. Van Roozendael, C. R. Nowlan, K. Chance, X. Liu, C. Lee, R. V. Martin, Application of OMI, SCIAMACHY, and GOME-2 satellite SO2 retrievals for detection of large emission sources, JGR Atmospheres, (2013), doi: https://doi.org/10.1002/jgrd.50826.

He, H. and Li, C. and Loughner, C. P. and Li, Z. and Krotkov, N. A. and Yang, K. and Wang, L. and Zheng, Y. and Bao, X. and Zhao, G. and Dickerson, R. R., SO2 over central China: Measurements, numerical simulations and the tropospheric sulfur budget, JGR, (2012), doi: https://doi.org/10.1029/2011JD016473.

Koukouli, M. E. and Balis, D. S. and Theys, N. and Brenot, H. and van Gent, J. and Hendrick, F. and Wang, T. and Valks, P. and Hedelt, P. and Lichtenberg, G. and Richter, A. and Krotkov, N. and Li, C. and van der A, R., OMI/AURA, SCHIMACHY/ENVISAT and GOME2/MetopA sulphur dioxide estimates; the cas of Eastern Asia, 'ATMOS 2015, Advances in Atmospheric Science and Application', Heraklion, Greece, June 2015 (ESA SP-375, November 2015)

Rix, M. and Valks, P. and Hao, N. and van Geffen, J. and Clerbaux, C. and Clarisse, L. and Coheur, P-F. and Loyola, D. and Erbertseder, T. and Zimmer, W. and Emmadi, S., Satellite Monitoring of Volcanic Sulfur Dioxide Emissions for Early Warning of Volcanic Hazards, IEEE Journal of Selected Topics in Applied Earth Observations and Remote Sensing, (2009), doi: https://doi/org/10.1109/JSTARS.2009.2031120.

Rix, M. and Valks, P. and Hao, N. and Loyola, D. and Schlager, H. and Huntrieser, H. and Flemming, J. and Koehler, U. and Schumann, U. and Inness, A., Volcanic SO2, BrO and plume height estimations using GOME-2 satellite measurements during the eruption of Eyjafjallajökull in May 2010, JGR, (2012), doi : https://doi.org/10.1029/2011JD016718.

Wang, Y., Wang, J., Tropospheric SO2 and NO2 in 2012–2018: Contrasting views of two sensors (OMI and OMPS) from space, Atmospheric Environment (2020), doi: https://doi.org/10.1016/j.atmosenv.2019.117214.